# CAN WE TALK MODELS INTO SEEING THE WORLD DIFFERENTLY?

**Paul Gavrikov**[1,2,3,4]   **Jovita Lukasik**[5]   **Steffen Jung**[2,6]   **Robert Geirhos**[7]
**M. Jehanzeb Mirza**[8]   **Margret Keuper**[2,6]   **Janis Keuper**[1,2]

[1] IMLA, Offenburg University   [2] University of Mannheim   [3] Tübingen AI Center
[4] Goethe University Frankfurt   [5] University of Siegen
[6] Max Planck Institute for Informatics, Saarland Informatics Campus
[7] Google DeepMind   [8] MIT CSAIL

## ABSTRACT

Unlike traditional vision-only models, vision language models (VLMs) offer an intuitive way to access visual content through language prompting by combining a large language model (LLM) with a vision encoder. However, both the LLM and the vision encoder come with their own set of biases, cue preferences, and shortcuts, which have been rigorously studied in uni-modal models. A timely question is how such (potentially misaligned) biases and cue preferences behave under multi-modal fusion in VLMs. As a first step towards a better understanding, we investigate a particularly well-studied vision-only bias - the texture vs. shape bias and the dominance of local over global information. As expected, we find that VLMs inherit this bias to some extent from their vision encoders. Surprisingly, the multi-modality alone proves to have important effects on the model behavior, i.e., the joint training and the language querying change the way visual cues are processed. While this direct impact of language-informed training on a model's visual perception is intriguing, it raises further questions on our ability to actively steer a model's output so that its prediction is based on particular visual cues of the user's choice. Interestingly, VLMs have an inherent tendency to recognize objects based on shape information, which is different from what a plain vision encoder would do. Further active steering towards shape-based classifications through language prompts is however limited. In contrast, active VLM steering towards texture-based decisions through simple natural language prompts is often more successful.

**URL:** https://github.com/paulgavrikov/vlm_shapebias

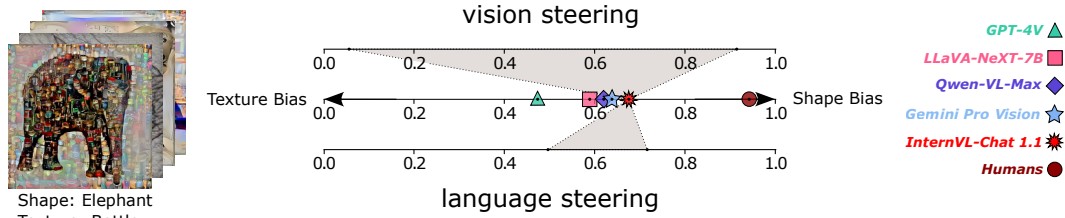

Figure 1: **Language can be used to steer visual cue preferences (biases) in vision language models (VLMs).** Here we illustrate the (visual) texture/shape bias (Geirhos et al., 2019) of some exemplary VLMs, and highlight the steerability of `InternVL-Chat 1.1` (Chen et al., 2024) through the processing of vision and text inputs (prompts).

# 1    INTRODUCTION

As the old adage goes, all models are wrong, but some are useful. Similarly, recent machine learning models have proven to be very useful in practice, although we know their decisions to be impacted by specific biases, such as cue preferences misaligned with human perception and shortcuts (Geirhos et al., 2020a). Some of these cue biases are particularly misaligned in traditional, uni-modal models and often reveal fundamental differences in the decision function compared to humans (Geirhos et al., 2019; Subramanian et al., 2024; Wang et al., 2020; Buolamwini & Gebru, 2018; Raji & Buolamwini, 2019). However, the current generation of deep learning models is increasingly multi-modal, for example, by fusion of large language models (LLMs) with modality-specific encoders (OpenAI, 2023; Alayrac et al., 2022; Huang et al., 2023). On the one hand, this approach allows for an exciting array of applications that can be defined at inference via prompts in natural language. On the other hand, the once well-studied biases are now combined in multi-modal fusion, leaving open questions on how and if the specific biases interact.

Specifically for vision language models (VLMs) we are therefore asking if we can *talk* models into *seeing* the world differently - *i.e.*, to what extent does the LLM-based multi-modal fusion change the cues predominantly used for image classification and, furthermore, can we utilize natural language prompts to override the inductive biases of vision encoders. If language is indeed able to influence a vision-only bias, this may offer the possibility of aligning model behavior (with human behavior) using intuitive language prompting.

In general, measuring biases in the visual cues used by a model to make a particular prediction is hard. While we assume that there is a multitude of cue biases learned in vision models, only a few of them are harmful or misaligned[1]. An example of a benign bias would be the "foreground" bias, *i.e.*, models mostly classify images by their foreground objects. Similarly, objects in the image center are usually perceived to be more important than objects in the periphery. These biases follow human intuition and have therefore not been causing much controversial discussion. This is in contrast to the texture vs. shape bias (Geirhos et al., 2019) - one of the best-studied cue biases in object recognition models (Hermann et al., 2020; Shi et al., 2020; Islam et al., 2021; Benarous et al., 2023; Naseer et al., 2021; Subramanian et al., 2024). It states that humans predominantly recognize objects in images by their shape (96% shape over texture decisions), whereas vision models strongly prioritize texture cues and discount the object's shape often. Machine perception is thus at odds with human intuition, even if the model accuracy is high.

We are the first to provide a large-scale study for VLMs, investigating the texture/shape bias in object recognition in visual question answering and image captioning. Our investigation shows that the texture bias is by default far less pronounced in VLMs than in most previously studied vision-only models. As shown in Fig. 1, VLMs decide by shape more often than by texture - albeit not matching the human shape bias (96%). Further, we find that by using biased instructions, we can steer the model output to some extent in both directions, toward a texture or shape bias. This demonstrates that visual biases in multi-modal models can be influenced by language, opening up an exciting new possibility of aligning model outputs using language prompts, without the need for retraining. In summary, our large-scale study offers the following findings:

- We show that VLMs preserve cue biases of their vision-encoders only to some extent, *e.g.*, they are yielding decisions that are more shape-biased than pure vision models, albeit not reaching human levels of the shape bias (Sec. 4.1).

- We show that uni-modal vision encoders generate a flexible representation that contains texture and shape cues. Ultimately, the language modality (through the LLM) tends to suppress either of the cues, such that objects are recognized purely on the grounds of either shape or texture (Sec. 4.2).

- We find that VLMs offer a unique opportunity to steer visual biases through language alone, stemming from the multi-modal fusion. For instance, we can steer the shape bias as low as 49% and as high as 72% through prompting alone without significantly affecting the accuracy (Sec. 5). This ability is not limited to the texture/shape bias (Sec. 5.2).

---

[1]We are explicitly focusing on biases in terms of low-level vision cues such as *shape* versus *texture* or *high frequency* versus *low frequency*. High-level biases, that may have a societal impact, are therefore explicitly excluded from this investigation. Please see Appendix O for a discussion.

## 2 RELATED WORK

Sparked by the success of vision language pretraining (Radford et al., 2021; Jia et al., 2021; Zhai et al., 2022b; Sun et al., 2023b), where features extracted from image-text paired data are aligned in a joint embedding space, recent VLMs added language modeling during training (Li et al., 2022; Yu et al., 2022; Li et al., 2023b), enabling models to reason about images. Subsequently, finetuning these VLMs on instruction-following data (Alayrac et al., 2022; Liu et al., 2023b; Luo et al., 2023; Dai et al., 2023; Huang et al., 2023), such as reinforcement learning from human feedback (RLHF) (Ouyang et al., 2022), enables users to prompt these models, easing their usability for humans. Resulting models are commercialized (OpenAI, 2023; Gemini Team, 2023; Qwen Team, 2024) or open-sourced (Liu et al., 2023b; Dai et al., 2023; Chen et al., 2024), and consequently become accessible to a wide range of users.

This success of vision language models calls for improving our understanding of the visual cues leveraged by VLMs and the degree to which these can be affected through language. In particular, the fact that vision models leverage cues different from the ones intuitively used by humans has been widely discussed. In our study, we focus on the texture vs. shape bias, as a particularly well-studied example in vision-only models. Humans primarily rely on shape information to recognize objects. This is in contrast to standard ImageNet-trained vision models, such as convolutional neural networks (CNNs), which are biased towards texture to make their classification decisions (Geirhos et al., 2019). Nonetheless, shape information can still be present in layers/latent space of the model before the classifier (Hermann et al., 2020; Islam et al., 2021). Prior research has shown that the texture bias of CNNs can be reduced in training (Geirhos et al., 2019; Lukasik et al., 2023; Li et al., 2021; Hermann et al., 2020; Geirhos et al., 2021; Gavrikov et al., 2023; Jaini et al., 2024; Gavrikov & Keuper, 2024). But the network architecture has a high influence, and vision-only ViTs (Dosovitskiy et al., 2021) were shown to be more shape-biased by default (Naseer et al., 2021), more human-like (Tuli et al., 2021), scalable by data size (Zhai et al., 2022a), and can be explicitly designed to separate shape and texture in their token space (Naseer et al., 2021). Jointly embedding vision and language in these networks (but not CNNs), through CLIP (Radford et al., 2021), further increases their shape bias in zero-shot classification (Geirhos et al., 2021). Yet, these models still do not reach human levels. The only known models to achieve such levels are image-generative classifiers (Jaini et al., 2024), which also combine vision and language in a different manner.

**Measuring Texture/Shape Bias.** A cornerstone of our analysis is the measurement of the texture/shape bias in (LLM-based) VLMs when performing tasks that are based on object recognition. In the following, we summarize how this bias is measured for vision-only models, which forms the basis for our study.

Like most studies on the texture/shape bias in vision models, we use the texture-shape cue-conflict classification problem (*cue-conflict*) (Geirhos et al., 2019) consisting of 1,280 samples with *conflicting* shape and texture cues synthetically generated via a style transfer model (Gatys et al., 2016) from ImageNet (Deng et al., 2009) samples (see Fig. 1 for examples). The shape and texture classes belong to 16 super-classes of ImageNet. Following (Geirhos et al., 2019), we have excluded 80 images from the dataset where texture and shape cues belong to the same class. From an information perspective alone, predicting either label (or both) would be correct. However, humans tend to prioritize the shape cue for predictions, which is in stark contrast to most models (Geirhos et al., 2019; 2021).

Using the shape or texture cue label as the correct label allows us to measure the *shape* and *texture accuracy*, respectively. Based on these measurements, we measure the *cue accuracy* as the ratio of predictions that contain either the shape or texture label (as opposed to a misclassification):

$$Cue\ Accuracy = Shape\ Accuracy + Texture\ Accuracy \qquad (1)$$

Throughout the paper, we will refer to this as the *accuracy*. We use the definition of *shape bias* (Geirhos et al., 2019), which is defined by the ratio of shape decisions over accurate decisions:

$$Shape\ Bias = Shape\ Accuracy / Cue\ Accuracy \qquad (2)$$

While we primarily focus on measuring the shape bias in this study, accuracy is an important signal for steering in later sections, as it demonstrates how the bias was obtained. For instance, achieving perfect shape bias in any model would be possible by simply mislabeling all texture-biased detections – but it would affect accuracy. Importantly, changes in accuracy (positive or negative) observed on the cue-conflict dataset may not necessarily generalize to other datasets.

## 3 MEASURING CUE BIASES IN VLMS

Given a dataset such as proposed in (Geirhos et al., 2019) for shapes and textures, we propose to measure the cue bias of VLMs in two tasks: *visual question answering (VQA)* (Antol et al., 2015), where we seek to obtain a zero-shot classification (Radford et al., 2021) of the object, and *image captioning* (Vinyals et al., 2015) where we look for an accurate but brief description of objects in the image. For both tasks, we evaluate single-round answering with no shared conversation history between conversations.

### 3.1 VQA CLASSIFICATION

Following the questioning style in `LLaVA` (Li et al., 2023a), we ask the model `"Which option best describes the image?"` and provide an alphabetic enumeration of all class labels in the style `"A. airplane"`. For a simpler response extraction and confidence evaluation (see below), we end the prompt by instructing the model to answer with only the letter corresponding to the correct answer (`"Answer with the option's letter from the given choices directly."`). Compared to captioning, this is similar to the discrimination in ImageNet (Deng et al., 2009) image classifiers (Krizhevsky et al., 2012; Srivastava et al., 2015; He et al., 2015; Huang et al., 2017; Dosovitskiy et al., 2021) in the sense that it only allows the model to respond with a single class and does not provide an option to not answer - if models follow the instruction.

**Response Extraction.** Despite instructing the models to only respond with an option letter, we observe multiple response styles: option letter + label (`"H. cat."`), just the label (`"cat."`), long explanation containing the option letter and/or label (`"The image features a black and white image of a cat."`). In all cases, punctuation and capitalization may be different (`"H.", "H", "h)"`). The first two response styles are easily correctable by simple post-processing (we prioritize the option letter in case of a conflicting option letter and label), and in some cases, explanations can be corrected as well if the response includes the option letter. However, we avoid heavy post-processing and consider individual answers wrong if they are not recoverable. In most cases, the ratio of these is negligible.

#### 3.1.1 ACTIVE STEERING THROUGH PROMPTS

The above setting allows us to test the inherent cue bias of a given VLM. Yet, the multi-modal nature of VLMs paves the way to not only test for a given bias but also to *actively steer* the model towards using particular types of visual cues. Note that models are not trained to perform well under this task, and it is unclear how flexible they are in basing a particular decision on one or another type of cue (for example on texture or shape).

To explore the flexibility of model predictions for given types of visual cues, we conduct experiments comparing performance under a default neutral prompt and biased instructions.

In the simplest case, we can test the cue bias steering through hand-crafted prompting, where a model is asked to identify the class using a particular visual cue (*e.g.*, `"Identify the primary shape in the image."`). Details on our tested biased prompts can be found in Appendix C.3.

**Automated Prompt Engineering.** To further enhance the steering signal provided by the language prompt, we further evaluate *automatically crafted prompts*. This is achieved by employing an LLM as optimizer (Yang et al., 2024) to continuously generate new prompts in natural language targeting to maximize either shape or texture bias in a feedback loop. We provide the LLM feedback about the achieved accuracy and shape bias. Additionally, we opt for greedy token sampling in the VLM to reduce noise in the feedback loop. For further details, we refer the reader to Appendix E.

### 3.2 IMAGE CAPTIONING

In this task, we are instructing models to generate brief descriptions (`"Describe the image. Keep your response short."`). We specifically request the model to provide a short response to encourage it to single out the most crucial aspects of the image according to its judgment. Additionally, this has the benefit of faster inference.

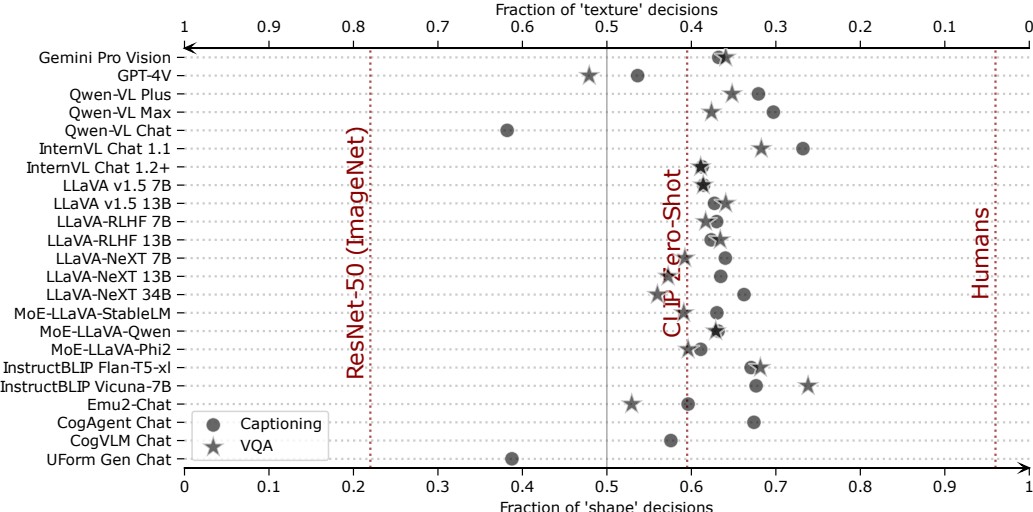

Figure 2: **Most VLMs prioritize shapes over texture cues.** We measure the shape bias on the `cue-conflict` dataset (Geirhos et al., 2019). For reference, we also provide measurements on an ImageNet-trained ResNet-50 (He et al., 2015), zero-shot classification with CLIP ViT-L/14 (Radford et al., 2021), and a human average (over 10 subjects (Geirhos et al., 2019)). The results in table format are shown in Appendix B.

**Response Extraction.** As the responses are open-ended, we rely on zero-shot classifications of the generated description to marginalize the most descriptive class. To this end, we embed the generated descriptions and all (raw) class labels using `ember-v1` (Nur & Aliyev, 2024) and predict the class with the smallest cosine distance (similar to zero-shot classification in CLIP (Radford et al., 2021)). However, the generated caption may refer to multiple class labels (or none). As an additional signal, we perform a more granular analysis using an additional LLM (`Nous-Hermes-2-Mixtral-8x7B-DPO` (Teknium et al., 2024)) by instructing the model to extract all mentioned classes (similar to Yan et al. (2021)). This allows us to understand if the model detects both cues (but the embedding model enforces a specific prediction) and to quantify how often the model response is too generic to detect any class.

## 4    CUE BIASES IN VLMs: AN ANALYSIS OF TEXTURE VERSUS SHAPE

We base our core analysis on the *texture/shape cue-conflict* dataset (Geirhos et al., 2019) evaluation of the texture versus shape bias. Traditional image classification models like ResNet-50 trained on ImageNet have been shown to severely prioritize texture cues (only 22% shape bias) starkly contrasting the strong shape bias of humans (96%). As of now, results on multi-modal LLM-based models are missing. Therefore, we start our experimental evaluation by measuring the shape bias for the VQA and Image Captioning tasks, using a collection of diverse VLMs reflecting the multitude of research directions. These models include connections of common pretrained CLIP encoders and LLMs (Liu et al., 2023b;a; 2024; Dai et al., 2023; Bai et al., 2023b; Hong et al., 2023; Wang et al., 2023; Sun et al., 2023a), mixture-of-expert-LLMs (Lin et al., 2024), optimized architectures for resource-constrained systems (Kim et al., 2023), finetuning with RLHF (Sun et al., 2023c; OpenAI, 2023), or massive vision encoders (Chen et al., 2024). Additionally, we survey commercial, closed-source models like `Gemini Pro Vision 1.0` (Gemini Team, 2023), `GPT-4V (Preview)` (OpenAI, 2023), and `Qwen-VL Plus/Max` (Qwen Team, 2024) where access is limited to APIs and few details are known. For a detailed list of models, please refer to Appendix A.

### 4.1    KEY RESULTS

The results in Fig. 2 paint a fairly uniform picture across different models and on two different tasks. Overall, the shape bias of VLMs is still significantly lower than that of humans (96%), but higher

Table 1: **Comparison between VLMs and their encoders.** We show the relative difference between a VLM and its encoder in shape bias and accuracy when evaluated on cue-conflict tasks, along with error consistency (Geirhos et al., 2020b). VLM performances are assessed using VQA, while the vision encoders are evaluated using zero-shot classification. Statistically significant changes in shape bias ($p < 0.05$ in a two-sided t-test) are denoted by an $*$ next to the value.

| VLM | Vision Encoder | VLM - Encoder [%] | | Error Consistency [%] |
|---|---|---|---|---|
| | | Accuracy | Shape Bias | |
| LLaVA v1.5 13B (Liu et al., 2023a) | ❄ CLIP ViT-L/14@336px (Radford et al., 2021) | −3.50 | +4.2* | 73.5 |
| LLaVA v1.5 7B (Liu et al., 2023a) | | −3.00 | +1.5 | 76.8 |
| LLaVA-NeXT 34B (Liu et al., 2024) | | −9.92 | −3.9* | 67.1 |
| LLaVA-NeXT 13B (Liu et al., 2024) | | −0.33 | −2.7 | 70.9 |
| LLaVA-NeXT 7B (Liu et al., 2024) | | −1.08 | −0.2 | 67.5 |
| MoE-LLaVA v1.5 Phi2 x4 (Lin et al., 2024) | | −1.42 | −0.3 | 73.4 |
| MoE-LLaVA v1.5 Qwen x4 (Lin et al., 2024) | | −24.25 | +3.0 | 51.4 |
| MoE-LLaVA v1.5 StableLM x4 (Lin et al., 2024) | | −3.67 | −0.8 | 73.1 |
| InstructBLIP FLAN-T5-XL (Dai et al., 2023) | ❄ EVA-01-CLIP ViT-g/14@224px (Sun et al., 2023b) | −6.83 | +1.8 | 73.7 |
| InstructBLIP Vicuna-7B (Dai et al., 2023) | | −14.42 | +7.4* | 78.7 |
| Emu2-Chat (Sun et al., 2023a) | ❄ EVA-02-CLIP-E/14+@448px (Sun et al., 2023b) | −11.08 | −9.5* | 61.0 |

than in typical image-only discriminative classifiers (*e.g.*, $22\%$ for an ImageNet-trained ResNet-50 (He et al., 2015; Geirhos et al., 2019)). Additionally, **for most models, the shape bias is higher than the ca. 60% shape bias of CLIP ViT-L/14** (Radford et al., 2021) - an interesting result given that this model is a common vision encoder used in many of our tested models. GPT-4V (OpenAI, 2023) is an unexpected outlier both in terms of accuracy and in terms of texture bias. We further discuss this particularity in Sec. 6.

**The task only marginally affects the shape bias.** Despite conceptually different tasks, *i.e.*, the discriminative VQA task and the one open-ended captioning task, we do not observe fundamental shifts in the utilized information cue. We were able to report shape bias under the image captioning task for all models. However, a few models did not follow the VQA instructions and are, thus, not reported in Fig. 2. Most of these models displayed a pronounced texture bias, which might hint towards a correlation between underfitting and texture bias, but to answer this question conclusively, we would need more samples.

On average, the shape bias is slightly higher for the image captioning task than for VQA (on average $63.9\%$ versus $61.6\%$ for those models that could be evaluated on both tasks). However, this comes at some cost in accuracy (on average $71.0\%$ versus $78.9\%$). This decrease in accuracy is due to generic captions that do not refer to any class (see Appendix B for details). For VQA the range is from $52.9$ - $73.8\%$ and $54.1$ - $73.2\%$ for captioning - yet, outliers with a significantly lower ($38.2\%$) shape bias in captioning exist and for the individual models, the cue bias strongly depends on the considered task (refer to the gap between circles and stars for several of the models in Fig. 2). Exceptions seem to be, for example, the Gemini Pro Vision 1.0 model, several of the LLaVA models, and the InternVL-Chat 1.2+ model for which the considered task barely influences the cue bias.

## 4.2 MECHANISTIC ANALYSIS

We have observed that the shape bias in VLMs differs from that of CLIP (ViT-L/14), the vision encoding model used in most of the tested models. This prompts an inquiry into the VLM's decision process. Specifically, it raises the question of whether language can affect the purely visual texture/shape bias. In this section, we want to specifically look at the vision encoder and its representation having only access to the visual input, and the LLM combining both modalities.

### 4.2.1 VISION ENCODING

Most VLMs combine a frozen CLIP vision tower with an LLM via some projector (Dai et al., 2023; Liu et al., 2023b; Alayrac et al., 2022). Hypothetically, the LLM could learn to perform zero-shot classification using their encoders akin to a function call whenever the prompt requires some form of classification and then simply forward the result. In such a case, the VLM would also inherit the shape bias from the encoder. To gain more insights, we ablate the encoder using zero-shot classification from the full model. We derive the encoder's predictions by calculating the cosine similarity between

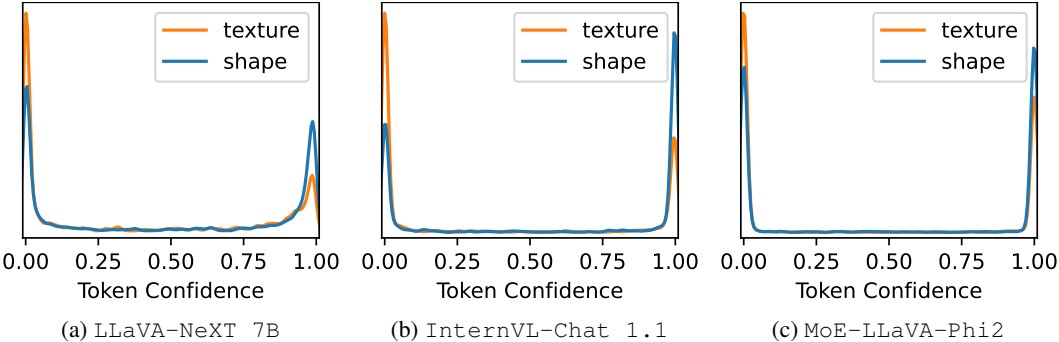

(a) `LLaVA-NeXT 7B`   (b) `InternVL-Chat 1.1`   (c) `MoE-LLaVA-Phi2`

Figure 3: **Confidence distribution of shape and texture tokens for all samples.** All models form highly biased decisions by completely ignoring one cue. Measured on `LLaVA-NeXT 7B`, `InternVL-Chat 1.1`, and `MoE-LLaVA-Phi2` for the VQA task.

the encoded class labels[2] and the input sample, selecting the label with the highest similarity to the image. Specifically, we measure the difference in accuracy and shape bias between the encoders of the VLM and the VLM itself, as well as their *error consistency* (Geirhos et al., 2020b).

Error consistency is a metric to assess whether two observers (*e.g.*, a human and a model) systematically make errors on the same images. If that is the case, this suggests a deeper underlying similarity compared to simply reaching similar overall accuracies, since those could be reached with very different strategies. The metric is based on Cohen's kappa (Cohen, 1960) and computes how frequently errors made by two observers overlap (up to perfect overlap) while correcting for the overlap expected by chance. Cohen's kappa is within $[-1, 1]$, with a value of $0$ indicating chance-level consistency, positive values indicating systematic agreement, and negative values indicating systematic disagreement. For our measurement, we treat all predictions other than the shape label that are not related to the shape cues as errors.

**The vision encoder provides a *flexible* representation.** Our comparisons in Tab. 1 show that VLMs' decisions differ from their isolated encoders. Even on the rather simple 16-way cue-conflict problem, all VLMs decrease in accuracy compared to their encoders in zero-shot classification. This may be expected as the increase in supported tasks in VLMs comes at a cost in specialization, but is already a sign of changes. Further, we note that the error consistency to their respective encoders only matches up to 78.7%. This leaves at least $20\%$ room for decisions that differ from the encoder, proving that the LLM and the text prompt further influence the shape bias, despite being a vision-only bias. This deviation can only be possible if the generated vision tokens are flexible to some degree by containing information belonging to both cues. This is further confirmed by the measurement of shape bias, which shows a $-9.5\%$ to $+7.4\%$ difference in **both** directions.

### 4.2.2   LLM PROCESSING OF VISION TOKENS

In the VQA task, we force the model to predict a single class - yet, the previous section has shown that the vision tokens generated by the encoder contain information from both cues. To better understand the processing, we evaluate the VQA prediction confidences as follows.

All answer options in our VQA prompts correspond to a single character and, thus, a token. Well-behaving models, where the response consistently starts with the option letter (and nothing else) allow us to gather insights into the prediction process. For these models, the logits of each token correspond to the logits of option letters. By applying a *softmax* function, we can analyze the confidence in each option (we map invalid tokens to a separate null class). This allows us to better study the sampling behavior and make the following observation:

**LLMs turn flexible representations into biased decisions.** In Fig. 3, we visualize the confidence of the token corresponding to the shape or texture answer option. This experiment can only be performed on models where we have access to the logits, and the model consistently follows the instructions to only respond with the predicted options letter. This limits the analysis to a few models

---

[2]We explore various prompt templates (and ensembles) in Appendix J, yielding consistent results.

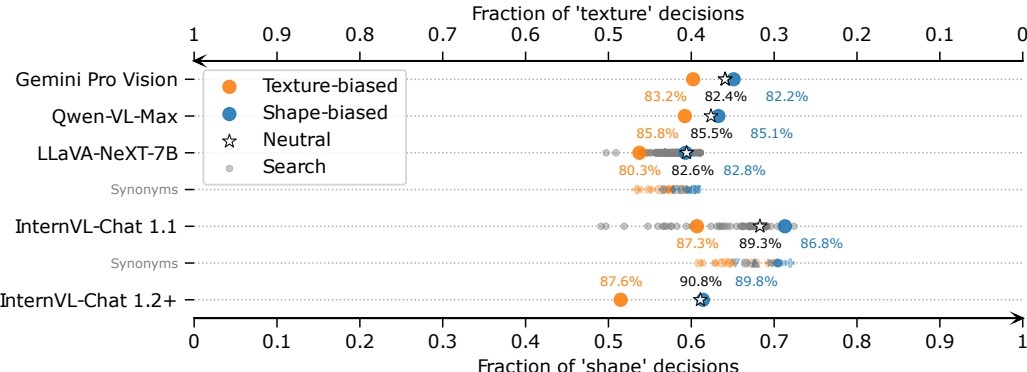

Figure 4: **Language can steer the texture/shape bias to some extent.** We test the same texture/shape-biased instructions on multiple models, showing that these can already shift some decisions (usually in favor of texture). The stated percentages refer to the achieved accuracy on cue-conflict. For `InternVL 1.1` and `LLaVA-NeXT 7B` we additionally test the understanding of texture/shape by using synonyms. Furthermore, we use an LLM to automatically search for specific prompts to optimize in either direction.

in our zoo, and we show similar results on `LLaVA-NeXT 7B` (Liu et al., 2024), `InternVL 1.1` (Chen et al., 2024) and `MoE-LLaVA-Phi2` (Lin et al., 2024).

To our surprise, we find that confidence in both options is almost binary. Analogously, when we only focus on correct answers, we observe that the model is highly confident in its responses. As the model places such high confidence in the selected cue, this suggests that information from the alternative cue is effectively disregarded during LLM processing. This conclusion is further supported by the observation that the second, significantly less confident prediction token does not align with the conflicting cue. For example, in `LLaVA-NeXT 7B` (Liu et al., 2024), this occurs in 70.7% of cases. Only 17.7% of the top-2 pairs contain both, shape and texture. Thus, while the encoder has its own inductive bias that directs the final decision, the actual biasing happens in the LLM, similar to linear classification heads in discriminative models (Islam et al., 2021).

While it is not clear if these findings generalize to all other VLMs, the overall high error consistency between all VLMs (see Fig. 13 for a heatmap) on our task hints that our results may generalize well to other models.

## 5 PROMPT-BASED STEERING OF VLM OUTPUTS

In the previous section, we have seen that in VLMs, visual biases are not simply inherited from the vision encoder, but the fusion with an LLM including the text prompt plays a crucial role. Given the somewhat flexible representation of texture/shape bias, we test in the following if we can actively talk VLMs into seeing the world differently, i.e., systematically *steer* the output towards either end of the bias and go beyond the inductive bias. Therefore, we explore the influence of language through prompt engineering on the visual bias (Sec. 3.1.1). Furthermore, since texture/shape bias is a vision-only bias, we contrast this language steering to visual steering through image preprocessing (see Appendix F). While we expect strong model steerability through image preprocessing (obviously at high costs in model accuracy), Fig. 1 indicates significant flexibility of (some) VLMs through language, potentially offering a powerful way of shaping visual biases in a user-specified way without the need to retrain a model.

### 5.1 STEERING TOWARDS SHAPE- OR TEXTURE-BIASED OUTPUTS

Our previous results suggest that VLMs learn a connected multi-modal understanding of shape and texture. This opens the question of whether visual biases in outputs can be influenced through text processing in these models. We test this hypothesis by recording texture/shape bias as a function, steering it via text through prompt engineering.

**Bias steering through hand-crafted prompting.** We start by asking VLMs to specifically identify either the "shape" or the "texture" category in a given cue-conflict image. As shown in Fig. 4, prompting can steer a visual bias (*without* significantly affecting accuracy). Details on these biased prompts can be found in Appendix C.3. Neutral prompts often perform similarly to shape-biased prompts, whereas texture-biased prompts deviate more significantly. This suggests that models may be more inclined to use shape by default, but also have access to a certain amount of texture information which can be accessed through biased prompting.

To better test the representation of "shape" and "texture", we additionally replace the terms texture/shape with strong synonyms obtained from *Thesaurus.com* (Dictionary.com, 2024a;b). Then we measure shape bias on (`InternVL-Chat 1.1` (Chen et al., 2024) and `LLaVA-NeXT 7B` (Liu et al., 2024)). Synonyms of either term can steer shape bias as well to a certain degree. For "texture" synonyms, we observe more variance, as "texture" is overloaded by different meanings (*e.g.*, some synonyms like "feeling", "taste", or "touch" are unrelated to texture in vision). In contrast, "shape" is a fairly well-defined term. This demonstrates that the steering is not coincidental but leverages a learned representation.

While the effect of steering by language is systematically visible, language steering alone does not fundamentally change the reliance on the underlying cue. This effect does not appear to be a limitation of LLM capacity - the evaluation of `InternVL-Chat 1.2+` (34B vs. 13B) does not provide evidence that larger LLMs offer more steerability.

**Bias steering through automated prompt engineering.** Did our results indicate a limit on how much language/prompting can influence biases, or merely reflect that the handcrafted prompts were chosen suboptimally? To address this question, we test *automatically crafted prompts* (see Sec. 3).

The results are shown in Fig. 4 in gray and denoted as "search". We observe that for both `LLaVA-NeXT-7B` and `InternVL-Chat 1.1`, automatically generated prompts exceed the manually crafted biased prompts in terms of their effectiveness to increase texture bias, and roughly match them when it comes to increasing shape bias. For `InternVL-Chat 1.1` the delta between both extremes is 23.3%, which can only serve as a lower bound and is likely improvable by a better design of the LLM task (or using other optimizers). In line with hand-crafted prompts, overall accuracy does not change considerably or sometimes even improves. We should also note that the optimization is done for the cue-conflict test set; this is simply done as a proof of concept to show that there are prompts that can influence visual biases substantially, and not to claim a SOTA shape bias.

## 5.2 STEERING TOWARDS LOW- OR HIGH-FREQUENCY-BIASED OUTPUTS

So far, we have analyzed the texture/shape bias. In this section, we show that steering is possible for other biases, too. To this end, we explore a bias originating in the spectral domain, specifically focusing on low versus high-**frequency cue conflicts** (Oliva et al., 2006). This bias has been shown to affect a classification model's robustness, for example in (Wang et al., 2020; Lukasik et al., 2023). A related observation has been made in (Subramanian et al., 2024), showing that the critical frequency band of object recognition separates human perception from model vision.

Table 2: **Prompt steering on frequency-cue-conflict.** Statistically significant changes are marked by * (two-sided t-test with $p < 0.05$). We compare the "neutral" prompt with found prompts to maximize ("search (max)") or minimize the respective bias ("search (min)").

| Model | Prompt | Accuracy [%] | LF Bias [%] |
|---|---|---|---|
| `InternVL-Chat 1.1` | neutral | 92.92 | 34.5 |
| | search (max) | 90.33 | 38.6* |
| | search (min) | 91.33 | 32.9 |
| `LLaVA-NeXT 7B` | neutral | 82.83 | 52.4 |
| | search (max) | 84.25 | 54.5 |
| | search (min) | 82.67 | 48.7* |

To study the steerability of the low-frequency bias (LF bias), we propose a new dataset of stimuli following the texture/shape cue-conflict benchmark methodology (Geirhos et al., 2019): We create 1,200 samples belonging to 16 ImageNet-super-categories by blending the spectral components of two differently labeled images: 30% of the low-frequency components from one image and 70% of the high-frequency components from the other. Please find the details Appendix G.

We test the same neutral prompt as for previous experiments and use our automated prompt search to either maximize or minimize the bias. The results in Tab. 2 show that again we can steer the bias by language. The range of course depends on the vision representation and language training and is not as pronounced as for texture/shape bias. Still, we find that our prompts can result in statistically significant changes in bias. On `LLaVA-NeXT 7B`, the prompts even improved accuracy alongside the bias.

Independent of the prompting, it is interesting that the smaller `LLaVA-NeXT 7B` model almost perfectly balances the conflicting cues, whereas the larger `InternVL 1.1` model is significantly biased toward HF. We hope that this dataset can pave a new avenue for future research on frequency bias.

## 6 CONCLUSION

In our study, we investigated the processing of visual cues in multi-modal models, specifically if language can be used to change the visual perception of these models.

We acknowledge that the broader research question extends beyond the scope of any single study. Thus, our work focuses on a specific, well-defined example of a visual bias: the texture/shape bias (Geirhos et al., 2019) and investigates how language can be used to favor texture or shape in their predictions. Furthermore, we extend our analysis beyond texture/shape bias to include evaluations of low/high-frequency bias, demonstrating that the steerability of a model's bias is not limited to a single characteristic.

Indeed, we were able to show that visual cues are influenced by language. Utilizing this finding, we can show an intriguing aspect of VLMs: biases in the response can be steered through simple natural language prompts offering a form of alignment. This unique trait differentiates LLM-based from task-specific and uni-modal models. While this form of steering was not able to fundamentally change the utilized cue for the evaluated biases[3], it comes at almost no impact in accuracy, and most importantly does not require any retraining of the model. In fact, many attempts at steering shape bias in training have yielded worse results through more expensive methods (Li et al., 2021; Lukasik et al., 2023). Instead, we provide a simple and intuitive way for practitioners to adjust the output beyond the inductive bias at runtime with minimal effort.

Taken together, we can indeed talk VLMs into seeing the world differently.

**Limitations.** Even though we utilized a diverse array of VLMs, there is a possibility that different models would lead to different conclusions. We believe our results provide a fair reflection of the current VLM landscape, but radically different VLM architectures may lead to changes in bias mechanics and consequently alter our findings. Further, not all models have been behaving as expected in our study: Given that `GPT-4V` often achieves SOTA performance and is considered an important baseline, it has surprisingly poor accuracy in both VQA and image captioning tasks compared to most other models - mostly due to refusal to answer which affected 131/1280 VQA conversations, *i.e.*, roughly 10%. This is substantially higher than the refusal rate of all other models ($< 1\%$). It is worth noting that refusal rates do not affect the shape bias measurement. `GPT-4V` is also the model with the largest amount of generic image captions (60.4%). Additionally, we acknowledge that other prompts may have led to better results, however, the result is noteworthy, as the other VLMs mostly behave well under the same prompts. Overall, prompting is a potential source of bias in our study. Different prompts could have yielded different results, and certain models might have performed differently, particularly in Visual Question Answering (VQA) tasks. While we mitigated this by utilizing simple, widely used prompts, other choices remain to be explored in future investigations. We provide a brief exploration of alternatives in Appendix C.2.

---

[3]Interestingly, our observed behavior of limited steerability has similarities to human perception experiments conducted by Geirhos et al. (2019). In their control experiments, humans were either instructed to identify the shape while ignoring the texture or conversely to identify the texture while ignoring the shape. This "human prompt steering" worked, but only to a certain extent: When humans were tasked to ignore the shape, the human shape bias decreased from 96% (neutral instruction) only to approx. 70% shape bias (texture-biased instruction). Our tested VLMs behave somewhat similarly: their visual shape bias can be steered through prompting, but it appears hard for them to completely go against their default visual bias.

ACKNOWLEDGMENTS

PG and JK acknowledge financial support by the *German Federal Ministry of Education and Research (BMBF)* in the program *"Forschung an Fachhochschulen in Kooperation mit Unternehmen (FH-Kooperativ)"* within the joint project *LLMpraxis* under grant 13FH622KX2. PG is additionally supported by the *German Federal Ministry of Education and Research (BMBF)* project *STCL - 01IS22067*. JL acknowledges support from the *Lamarr Institute for Machine Learning and Artificial Intelligence*. JL, SJ, and MK acknowledge support by the *German Research Foundation research unit 5336 "Learning to Sense"*.

We thank Priyank Jaini for his valuable feedback.

REPRODUCIBILITY STATEMENT

We used open-source data and models, with detailed descriptions of our evaluation process provided in the appendix. The source code, evaluation results containing model answers for each sample, prompts generated by the automated prompt search, and the frequency-cue-conflict dataset, along with its creation scripts, are available at: https://github.com/paulgavrikov/vlm_shapebias.

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

APPENDIX

## A  OVERVIEW OF VLMS

Here we provide an overview of all used models from the main paper.

| | |
|---|---|
| `Qwen-VL-Chat` (Bai et al., 2023b) | Adds vision capabilities to `Qwen-7B` (Bai et al., 2023a). We set a repetition penalty of 1.2 for this model. |

| | |
|---|---|
| `Qwen-VL Plus/Max` (Qwen Team, 2024) | AliBaba's proprietary larger variants of `Qwen-VL-Chat`. Access only via API. |
| `CogAgent` (Hong et al., 2023) | A special model for interaction with graphical user interfaces (GUIs) at high-resolution. |
| `CogVLM` (Wang et al., 2023) | Adds "trainable visual expert module" in LLM layers to combine vision and language. |
| `Emu2` (Sun et al., 2023a) | The 37B model claims "strong multi-modal in-context learning abilities". |
| `InstructBLIP` (Dai et al., 2023) | Connects frozen vision encoders and LLMs through a trainable Q-Former. Uses `Vicuna` or `FLAN-T5` as LLMs. |
| `LLaVA v1.5` (Liu et al., 2023a) | Improvements of `LLaVA` with modifications on the image encoder, the projector, and task-specific data. Uses `Vicuna-7/13B` as LLM. |
| `LLaVA-NeXT` (Liu et al., 2024) | Successor of `LLaVA v1.5` supporting higher resolutions through patching, and using better SFT training data for training, claiming *"improved reasoning, OCR, and world knowledge"* (Liu et al., 2024). The 34B version switches from `Vicuna-7/13B` to `Nous Hermes 2 Yi 34B`. |
| `MoE-LLaVA v1.5` (Lin et al., 2024) | Variants of `LLaVA v1.5` employing 4 sparsely activated Mixture-of-Experts (MoE), and smaller LLMs (`Qwen`, `Phi-2`, `StableLM`). |
| `LLaVA-RLHF` (Sun et al., 2023c) | Variants of `LLaVA v1.5` aligned with Factually Augmented RLHF (Fact-RLHF) (Sun et al., 2023c). |
| `UForm-Gen Chat` (Kim et al., 2023) | A small (1.5B) model for VQA and image captioning finetuned for multi-modal chat. |
| `Gemini 1.0 Pro Vision` (Gemini Team, 2023) | Google's proprietary multi-modal model based on the Gemini Pro LLM. Access only via API. |
| `InternVL Chat 1.1/1.2+` (Chen et al., 2024) | An open-source effort to provide an alternative to `ViT-22B` (Dehghani et al., 2023). V1.1 is based on a 6B ViT and `Vicuna-13B`, V1.2+ uses `Nous Hermes 2 Yi 34B` as LLM including additional SFT on 10x more data. |
| `GPT-4V (Preview)` (OpenAI, 2023) | OpenAI's proprietary multi-modal model based on the GPT-4 LLM. Access only via API. Often considered to be the most powerful model. |

For our main analysis, we prompt all models at the default generation parameters (*e.g.*, temperature) unless stated otherwise. Appendix H shows why this is not an issue.

## B    DETAILED RESULTS TABLE

Tab. 4 shows the shape bias and accuracy for the VQA and Image Captioning task (see Fig. 2 in the main paper for a visualization). For the open-ended Image Captioning responses, we additionally provide evaluations through an LLM (see Sec. 3). These include the number of generated tokens (to measure how effective our "Keep [...] short." instruction is), the ratio of responses where exactly one class was detected (*single class ratio*), and the ratio of responses that do not refer to any description (*generic ratio*). In Appendix L/Tab. 9, we provide ablations on Image Captioning under the removal of generic responses.

Further, we make the following observations:

**Which models are the most shape-biased?**    The strongest shape bias is observed in `InstructBLIP Vicuna-7B` (Dai et al., 2023) for VQA, but the model generally shows a lower

accuracy compared to other models. A more accurate model is `InternVL-Chat 1.1` (Chen et al., 2024) which ranks second place for VQA but first for captioning.

**Does LLM scale matter?**   LLM capacity does not seem to correlate with shape bias and unpredictably skews the shape bias by a few percent in each way as can be seen in `Qwen-VL`, `LLaVA v1.5/NeXT/RLHF`, or `InternVL`. Similarly, the overall largest models do not have the highest shape bias. However, following the overall general trend, in our experiments, we also found that scale usually improves accuracy.

**Does RLHF align shape bias?**   RLHF-tuned VLMs are still rare at this point and we only have three samples. On both `LLaVA-RLHF` (Sun et al., 2023c) models we see no changes in comparison to the default LLaVA models. `GPT-4V` (OpenAI, 2023) (though it is unclear if vision was also RLHF trained) shows one of the lowest shape biases in our study, but we do not know how the base model ranks. Overall it is hard to derive a conclusive answer, but it seems that at least RLHF does *not necessarily guarantee* an alignment of visual preferences.

Table 4: The shape bias and respective accuracy on the cue-conflict dataset for various VLMs in VQA classification or image description tasks. For the image description task, we additionally provide the average number of tokens generated by `Vicuna`'s tokenizer and the ratio of responses that only contain a single class or are generic (do not mention any class) as judged by a separate LLM. "-" indicates models that did not follow instructions on VQA and could, thus, not be evaluated.

| | VQA | | Image Captioning | | | | |
| Model | Shape Bias [%] | Accuracy [%] | Shape Bias [%] | Accuracy [%] | Avg. Tokens | Single Class Ratio [%] | Generic Ratio [%] |
|---|---|---|---|---|---|---|---|
| Gemini 1.0 Pro Vision (Gemini Team, 2023) | 64.1 | 82.33 | 63.2 | 68.00 | 18.9 | 63.0 | 32.3 |
| GPT-4V (Preview) (OpenAI, 2023) | 47.9 | 69.75 | 53.6 | 52.67 | 44.8 | 37.2 | 60.4 |
| Qwen-VL Plus (Qwen Team, 2024) | 64.8 | 82.92 | 67.9 | 65.50 | 21.9 | 59.2 | 36.0 |
| Qwen-VL Max (Qwen Team, 2024) | 62.4 | 85.50 | 69.7 | 68.50 | 151.9 | 52.1 | 41.0 |
| Qwen-VL Chat (Bai et al., 2023b) | - | - | 38.2 | 67.42 | 27.3 | 59.1 | 33.2 |
| InternVL Chat 1.1 (Chen et al., 2024) | 68.3 | 89.33 | 73.2 | 75.58 | 16.9 | 74.9 | 19.4 |
| InternVL Chat 1.2+ (Chen et al., 2024) | 61.1 | 90.83 | 61.3 | 82.42 | 15.8 | 80.4 | 11.4 |
| LLaVA v1.5 7B (Liu et al., 2023a) | 61.4 | 80.75 | 61.4 | 76.08 | 12.1 | 73.8 | 19.2 |
| LLaVA v1.5 13B (Liu et al., 2023a) | 64.1 | 80.25 | 62.7 | 75.58 | 28.9 | 65.8 | 23.8 |
| LLaVA-RLHF 7B (Sun et al., 2023c) | 61.7 | 68.08 | 63.0 | 71.83 | 47.9 | 65.1 | 24.7 |
| LLaVA-RLHF 13B (Sun et al., 2023c) | 63.4 | 80.42 | 62.3 | 73.25 | 38.3 | 64.7 | 27.7 |
| LLaVA-NeXT 7B (Liu et al., 2024) | 59.2 | 82.58 | 64.0 | 65.08 | 20.2 | 55.5 | 39.5 |
| LLaVA-NeXT 13B (Liu et al., 2024) | 57.2 | 83.42 | 63.5 | 65.25 | 48.8 | 52.6 | 40.9 |
| LLaVA-NeXT 34B (Liu et al., 2024) | 56.0 | 73.83 | 66.2 | 57.50 | 93.4 | 36.2 | 59.1 |
| MoE-LLaVA-StableLM (Lin et al., 2024) | 59.1 | 80.08 | 63.0 | 73.92 | 24.1 | 67.4 | 21.6 |
| MoE-LLaVA-Qwen (Lin et al., 2024) | 62.9 | 59.50 | 63.2 | 75.33 | 13.3 | 69.4 | 20.7 |
| MoE-LLaVA-Phi2 (Lin et al., 2024) | 59.6 | 82.33 | 61.1 | 75.42 | 34.9 | 67.0 | 18.6 |
| InstructBLIP Flan-T5-xl (Dai et al., 2023) | 68.2 | 79.58 | 67.1 | 81.50 | 116.7 | 57.0 | 22.3 |
| InstructBLIP Vicuna-7B (Dai et al., 2023) | 73.8 | 72.25 | 67.7 | 80.67 | 94.0 | 60.9 | 28.0 |
| Emu2-Chat (Sun et al., 2023a) | 52.9 | 75.08 | 59.6 | 65.00 | 13.6 | 63.0 | 34.0 |
| CogAgent Chat (Hong et al., 2023) | - | - | 67.4 | 60.33 | 40.1 | 49.6 | 47.7 |
| CogVLM Chat (Wang et al., 2023) | - | - | 57.6 | 66.58 | 35.8 | 53.2 | 40.1 |
| UForm Gen Chat (Kim et al., 2023) | - | - | 38.8 | 64.50 | 30.2 | 59.3 | 33.0 |

# C   OVERVIEW OF PROMPTS

This section provides an overview of all the prompts we have used in our study, including fine-grained details and ablation studies on their effectiveness.

## C.1   MAIN PROMPTS

The prompt for VQA Classification is:

```
"{VQA_INSTRUCTION}
A. airplane
B. bear
C. bicycle
D. bird
E. boat
F. bottle
```

```
G. car
H. cat
I. chair
J. clock
K. dog
L. elephant
M. keyboard
N. knife
O. oven
P. truck
Answer with the option's letter from the given choices directly."
```

with a default setting `VQA_INSTRUCTION="Which option best describes the image?"`.

We use the following prompt for the Image Captioning task: `"Describe the image. Keep your response short."`

## C.2 EXPLORATION OF ALTERNATIVE PROMPTS

In initial testing, we found that the choice of prompts affects the eventual results and has the potential to inevitably influence our study. Thus, in an effort to address this, we extensively evaluated our models with multiple different prompting techniques, used in literature (Liu et al., 2023a; Dai et al., 2023) and chose the best one. Our prompt for VQA is inspired by LLaVA's prompts for multiple-choice questions [4]. In an additional experiment (Tab. 5), we ablated alternative prompts on `LLaVA-NeXT 7B` (Liu et al., 2024). We change or use an empty `VQA_INSTRUCTION` and change options to CLIP-style options (`"X. a photo of a {class}"`). However, we only observed a minor fluctuation in accuracy and shape bias and no significant effects. Our default prompt delivers the best accuracy and is, thus, our preferred choice.

Table 5: Exploration of alternative VQA prompts.

| Prompt | Shape Bias [%] | Accu-racy [%] |
|---|---|---|
| `"Which option best describes the image? [...]"` (default) | 59.2 | 82.58 |
| Default with CLIP-style options | 59.5 | 81.92 |
| `"Describe the object in the image: [...]"` | 60.2 | 81.33 |
| `"Describe the object in the image: [...]"` with CLIP-style options | 59.4 | 80.17 |
| Empty instruction (just options) | 59.5 | 81.33 |

Our image captioning prompt is a reformulation of the VQA prompt ("Which option best describes the image?" → "Describe the image."). In the following, we ablate if the suffix ("Keep your response short.") may have interfered with our results. Additionally, we tested an alternative suffix that explicitly asks for more details on `LLaVA-NeXT 7B` (Liu et al., 2024). The results for the former investigation in Tab. 6, show that our suffix indeed did not heavily bias the results in terms of shape bias. While adding the suffix leads to an impact in accuracy, it reduces the ratio of generic descriptions (not referring to any class) and has on average almost 4x fewer tokens resulting in significantly faster inference. Switching the suffix to `"Be precise."` increases shape bias, but at the same time also increases the number of generated tokens and worryingly the ratio of generic responses. Overall, we find that captioning prompts are more fragile, but our chosen default prompt provides an intriguing balance. For all ablated prompts, we find that the shape bias is higher than in VQA.

---

[4] https://github.com/haotian-liu/LLaVA/blob/main/docs/Evaluation.md [Online; accessed 6. Mar. 2024]

Table 6: Exploration of alternative Image Captioning prompts.

| Prompt | Shape Bias [%] | Accu-racy [%] | Avg. Tokens | Generic Ratio [%] |
|---|---|---|---|---|
| `"Describe the image. Keep your response short."` (default) | 64.0 | 65.08 | 55.5 | 39.5 |
| `"Describe the image."` | 63.6 | 68.25 | 202.9 | 46.8 |
| `"Describe the image. Be precise."` | 67.3 | 64.50 | 166.2 | 50.6 |

## C.3 BIASED PROMPTS

**Hand-crafted.** For our hand-crafted biased prompts, we set `VQA_INSTRUCTION` `="Identify the primary {BIASED_TERM} in the image."`, with `BIASED_TERM="shape"` and `BIASED_TERM="texture"`, for shape-, and texture-biased prompts, respectively.

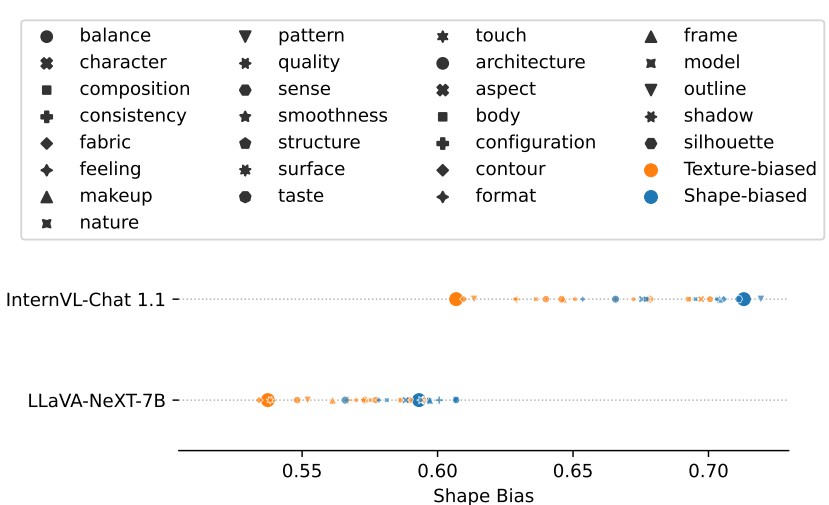

Figure 5: Detailed shape bias measurements under synonyms for biased VQA prompts.

**Synonyms.** We have retrieved the following strong synonyms from *Thesaurus* (Dictionary.com, 2024b;a).

- **shape:** architecture, aspect, body, configuration, contour, format, frame, model, outline, pattern, shadow, silhouette
- **texture:** balance, character, composition, consistency, fabric, feeling, make-up, nature, pattern, quality, sense, smoothness, structure, surface, taste, touch

The prompt is formed by replacing `BIASED_TERM` in the biased VQA prompt with the corresponding synonym. We did not filter out any synonyms but still want to emphasize that some of them are not reasonable in the context of vision (*e.g.*, for texture) which explains why the variance between them can be high. In Fig. 5, we also show the detailed shape bias measurements for each synonym (complementary to Fig. 4 in the main paper). We find that "pattern" - a synonym for both terms - tends to be more correlated with texture.

## D LLM-BASED RESPONSE EXTRACTION FOR CAPTIONS

We rely on an LLM to extract labels from the generated captions for the Image Captioning task. For every generated description we instruct `Nous-Hermes-2-Mixtral-8x7B-DPO` (Teknium et al., 2024) with the following prompt:

```
"Your task is to extract all objects that are described in the gi-
ven message. Only answer with all letters from the given choices
that apply. If none apply, reply with X. Do not explain. These are
the possible objects:
A. airplane
B. bear
C. bicycle
D. bird
E. boat
F. bottle
G. car
H. cat
I. chair
J. clock
K. dog
L. elephant
M. keyboard
N. knife
O. oven
P. truck
Message: {Generated Image Caption}"
```

Then we simply split the generated string into a list. We found this prompt by manually testing some examples and picking the best-performing one. For example, we experimented with other options to denote generic responses like "-". However, we found that this increases hallucinations, presumably as "-" is often used to begin bullet points and, thus, causes the model to continue generation.

## E    DETAILS ON AUTOMATED PROMPT SEARCH

Loosely inspired by (Yang et al., 2024), we utilized an LLM to optimize prompts. we switched to `Mixtral-8x7B-Instruct-v0.1`[5], as it performed better than the Nous Hermes version in early tests. The results shown in Sec. 5.1/Fig. 4 are a summary of multiple prompts that we tried in numerous multi-round conversations. In Fig. 6 we show the impact on accuracy: in most cases the accuracy remains similar or even improves. Of course, outliers where accuracy severely decreases exist.

We instruct the model to provide a prompt in a new line starting with "PROMPT: " that we then extract and automatically evaluate. Afterwards, we return the results to the LLM and ask it to generate the next prompt.

We have experimented with multiple prompts but ultimately our approaches can be loosely divided into prompts that try to *maximize* or *minimize* shape bias without significantly affecting accuracy. Besides linguistic tweaks, we experimented with the following techniques:

1. **Offering rewards:** We offered tips to the LLM to encourage it to generate more and better results[6]. However, Mixtral seems to be fine-tuned to refuse such attempts.

2. **Adding in-context examples**: We added an example (in language) of what it means to be shape or texture-biased in classification. This often seemed to bias the model to generate prompts that contain the example, too.

3. **Summarizing previous attempts**: We encouraged the LLM to summarize previous attempts before generating the next prompt, hoping to keep the most important aspects in context. The LLM did not always follow this suggestion.

---

[5] https://huggingface.co/mistralai/Mixtral-8x7B-Instruct-v0.1 [Online; accessed 6. Mar. 2024]

[6] https://twitter.com/voooooogel/status/1730726744314069190 [Online; accessed 6. Mar. 2024]

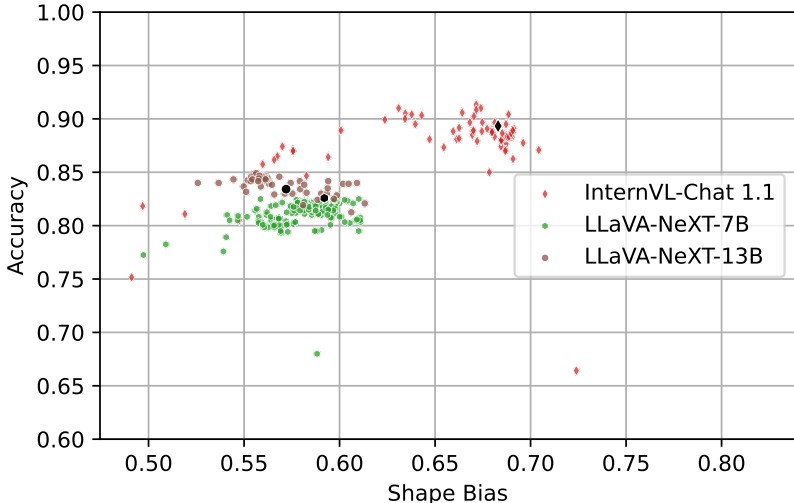

Figure 6: Shape bias vs. accuracy on cue-conflict in comparison for LLM generated prompts. Black points denote the default instruction.

4. **Returning the extracted prompt:** The LLM sometimes did not start the prompt with the requested prefix or misplaced it. We mitigated this by including the extracted prompt in our responses.

5. **Encouragements in response:** Initially, we only returned the accuracy and shape bias but found that the LLM sometimes abruptly quits the search. Thus we included encouragements in the form of questions like "What is your next prompt?". This seemed to improve conversations in terms of length, but could not entirely prevent the LLM from quitting.

6. **Simple but creative prompts:** When just instructed the LLM to generate prompts, we noticed that it would sometimes collapse to verbose prompts where it would attempt to rephrase terms by synonyms. Inspired by regularization terms in optimization, we ask the model to keep its prompt simple and creative to avoid minor tweaking in favor of more radical changes.

In all cases, we append a mock conversation (*i.e.*, both roles are written by us) to the history containing the neutral prompt and the respective shape bias/accuracy. An example conversation is shown in Tab. 11.

Optimization with LLMs is a highly exciting but also very active research field, where best practices have not yet emerged. For example, we have noticed that our instruction sometimes caused the LLM to refuse to continue when it found that the search was exhausted, or caused the LLM to maximize shape bias despite the instruction to minimize it. Overall, this is not an issue for our study as we are merely interested in understanding if quantitatively more texture/shape-biased prompts exist. The prompt shown in Tab. 11 (first message) is the final iteration integrating all of the above techniques.

# F  STEERING THE TEXTURE/SHAPE BIAS IN VISION

Earlier work demonstrated that ImageNet-models can still detect objects even if the image is split into patches and shuffled (Zhang & Zhu, 2019; Shi et al., 2020; Naseer et al., 2021). As patch size decreases, the operation is destroying more global shape information, yet retaining local texture information. We utilize this technique to significantly increase *texture bias*. Oppositely, to increase the *shape bias* we experiment with added Gaussian noise to inputs. This is loosely inspired by applying "diffusion-like noise" during training (and inference) which has been shown to drastically improve the shape bias of ImageNet-ResNets (Jaini et al., 2024). However, we only apply the noise during inference and use a more simplistic approach by adding $\mathcal{N}(0, \sigma^2)$ noise to all channels, consecutively clamping values to $[0, 1]$. We visualize the effects on one cue-conflict sample in Fig. 7.

Patch shuffling increases texture bias

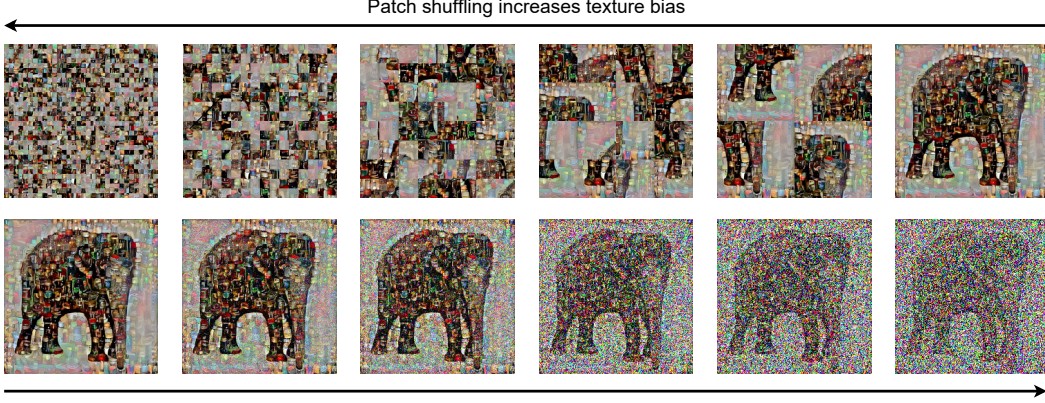

Noise increases shape bias

Figure 7: **Steering by Vision.** For one example image, we show how patch shuffling (top) increases texture bias by destroying shape information. Below we show how adding Gaussian noise increases shape bias by destroying texture information. Please note that we show more extreme values than those used in our experiments for visualization purposes.

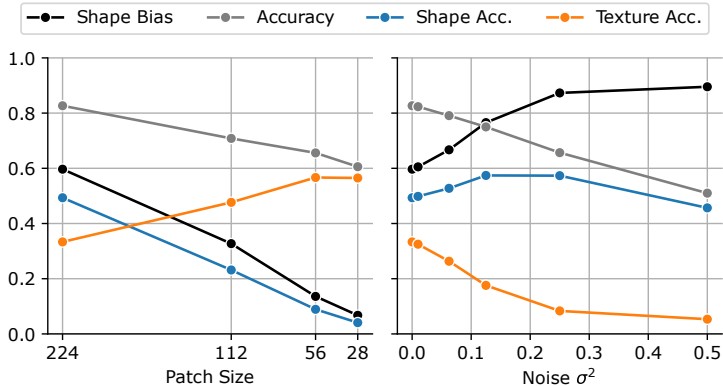

Figure 8: **Image preprocessing can strongly steer texture/shape bias.** Left: Shuffling image patches with decreasing patch size results in a strong texture bias. Right: Increasing Gaussian noise introduces a strong shape bias.

We show results on `LLaVA-NeXT 7B` in Fig. 8. Adding noise results increases the VLMs shape bias up to $89.5\%$ at $\sigma^2 = 0.5$, and patch shuffling decreases shape bias (increases texture bias) to $8.4\%$ at $28 \times 28$ patches. In both cases, the bias is indeed *steered* up to a certain threshold: the accuracy on one cue (texture or shape accuracy) increases whereas the accuracy on the other decreases. Beyond a specific point, the operation is destroying one cue entirely and can no longer be considered steering. Further, this form of steering comes at a cost in accuracy - yet, all results are still well beyond random chance. Inspired by these strong results, we repeat the experiments on the naturally more shape-biased and larger `InternVL-Chat 1.1` (Chen et al., 2024). In this model, we can further extend the range to $91.7\%$ shape bias ($\sigma^2 = 0.3$), and down to $6.1\%$ ($28 \times 28$ patches).

## G   FREQUENCY-CUE-CONFLICT

We create 1,200 samples belonging to 16 ImageNet-super-categories. The stimuli are generated by blending the low and high-frequency components of two differently labeled images. Specifically, we select two random ImageNet samples from the 16-class subset (Geirhos et al., 2019) with conflicting labels. Each image is then converted to grayscale by selecting the L channel, resized to 256 px on the shortest edge while preserving the aspect ratio, and finally center-cropped to $224 \times 224$ px. We

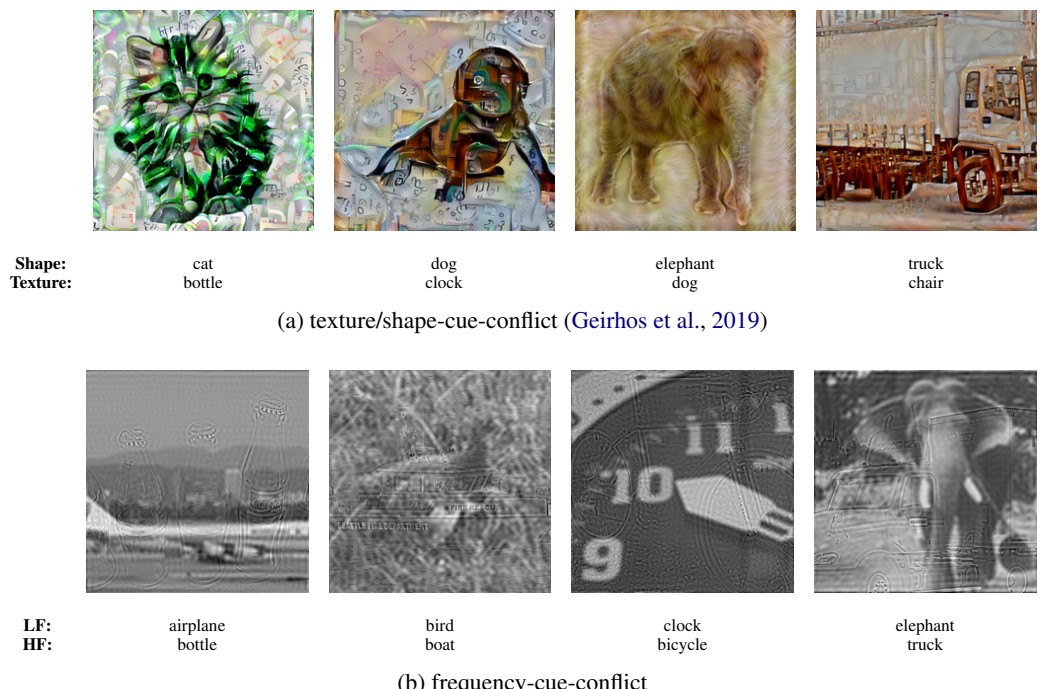

(a) texture/shape-cue-conflict (Geirhos et al., 2019)

(b) frequency-cue-conflict

Figure 9: **Examples from the cue-conflict datasets.** Images are constructed from two conflicting features cues: **(a)** texture/shape-cue-conflict (Geirhos et al., 2019) conflicting shape and texture; **(b)** frequency-cue-conflict conflicting high- (HF) and low-frequency (LF). Please zoom in for details.

finally blend the two images by using 30% of the low-frequency components from one image and 70% of the high-frequency components from the other. The resulting stimuli are saved as JPEG with 100% quality. Fig. 9b shows some examples from the dataset.

## H    ABLATION OF TEMPERATURE SCALING

We are interested in determining if generation parameters can influence the behavior of shape bias. Generally, VLMs only expose a few controllable parameters but all offer some form of stochastic sampling of tokens, often via *temperature scaling* of the token logits. Most models default to low-temperature settings (or settle for a greedy token strategy) which is more correlated with precise answers and reasonable for VQA. On the contrary, higher temperatures are correlated with more creative outputs and eventually token gibberish at extreme values. In general, temperature scaling also results in better-calibrated models (Guo et al., 2017).

Exemplarily, we study this on LLaVA-NeXT 7B for both VQA and Image Captioning. We repeat the non-greedy experiments 3 times for statistically meaningful results; however, we generally notice a marginal error between runs. Our results in Fig. 10, show no correlation between temperature and shape bias. As expected, the accuracy (slightly) decreases, because less confident tokens mapping to correct predictions are now replaced by false predictions. Yet, this affects texture/shape information alike. This can easily be explained by our token sampling analysis in Sec. 4. On average, texture/shape options are fairly similarly confident and top-1 tokens have a very high confidence (in the VQA setting) which the temperature scaling barely affects.

On the one hand, this finding serves as important confirmation that our comparison of VLMs at default values (picked by the original authors) is reasonable as it does not interfere with the shape bias. On the other hand, this implies that users seeking more creative outputs can tune the temperature (and similarly other parameters that control stochastic token sampling) without changing the underlying reasoning paths for vision inputs.

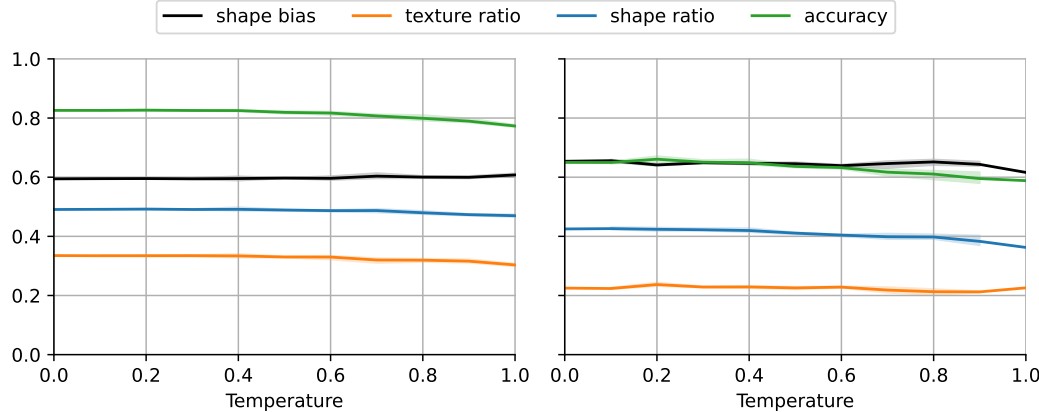

Figure 10: **Temperature scaling has no significant effect on shape bias neither under VQA (left) nor Image Captioning (right) tasks but starts to decrease accuracy at higher levels.** Experiments performed on `LLaVA-NeXT 7B` with 3 seeds (except Temperature = 0 and Temperature = 1 of Image Captioning where we use a single seed).

## I    AGREEMENT SETS ON TEXTURE/SHAPE CUE-CONFLICT

In this section, we aim to identify "agreement sets" of VLMs predictions on the texture/shape cue-conflict dataset (under the default VQA), *i.e.*, we want to find samples where all 19 models behave the same to see if we can identify some common patterns. Specifically, we search for samples where all models predict the shape label, texture label, or generally make a correct or wrong prediction, respectively. For each agreement set, we show 4 random examples in Fig. 11. For this analysis alone, it is not evident to us when models predict *texture* or *shape* labels, respectively. Overall, the very low number of samples where all models make a false prediction (6 out of 1,200) is a good indication of the quality of the texture/shape cue-conflict dataset (Geirhos et al., 2019). We find that these error samples are still recognizable (but we may be biased, knowing the ground truth). There seems to be also a quite large subset of samples where all models make correct predictions (320 out of 1,200).

## J    RESULTS ON CLIP MODELS

In this section, we provide results for CLIP models that we referenced in the main paper. We provide results for different architectures under three different prompting strategies: a computation of zero-shot centroids from 80 different prompts including usage of the class name (Radford et al., 2021), "a photo of {class}." which is often used as a default prompt (note the dot), and "{class}" (without dot). We will argue that the latter is more comparable to the VQA task of our VLMs - but of course, VLMs may have a better representation in weights. Either way, the shape bias does not significantly deviate between the three strategies. Tab. 7 shows the obtained shape bias (and accuracy) measurements.

We also noticed that the observed scaling laws in (Geirhos et al., 2021) do not always hold for vision encoders, despite an increase in parameters from EVA02-CLIP-E/14+ (5B) to EVA02-CLIP-8B, we actually see a significant decrease in shape bias (but an improvement in accuracy).

Our results also contain (rather uncommon) ResNet-based CLIP models. Note these are the only models, where the 80 prompts significantly improve accuracy. In terms of shape bias, ResNet-based CLIPs significantly underperform any ViT or ViT-based CLIP.

## K    RESULTS ON IMAGENET-TRAINED MODELS

Complementary to the results on CLIP, we also provide some shape bias evaluations of ImageNet-trained/finetuned models in Tab. 8. Note how ViTs are much more shape-biased than ResNets, as shown in (Geirhos et al., 2021), and yet, after ImageNet-finetuning, the previously well-performing CLIP (ViT-L/14@336px) model drops from 59.80 % to just 32.1 % of shape bias.

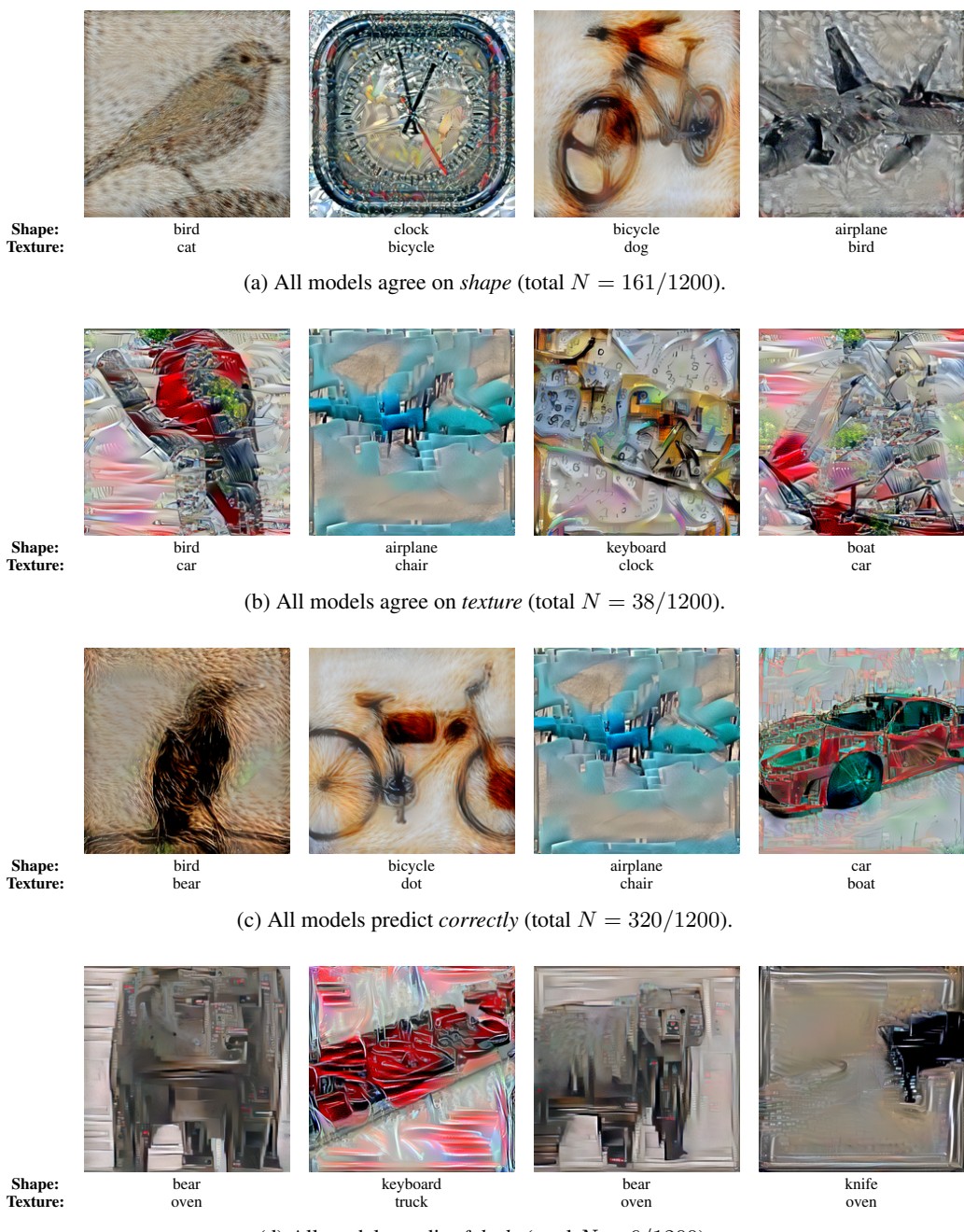

Figure 11: **Agreement sets.** We show 4 random samples where all VLMs show the same behavior under the default VQA prompt.

Table 7: Zero-shot classification on cue-conflict with different CLIP(-like) joint embedding models.

| Model | Prompt | Shape Bias [%] | Accuracy [%] |
|---|---|---|---|
| EVA01-CLIP-g/14 (Sun et al., 2023b) | 80 Prompts (Radford et al., 2021) | 66.03 | 87.83 |
| EVA01-CLIP-g/14 (Sun et al., 2023b) | "a photo of a {class}." | 66.03 | 87.08 |
| EVA01-CLIP-g/14 (Sun et al., 2023b) | "{class}" | 66.44 | 86.67 |
| EVA02-CLIP-8B@448px (Sun et al., 2024) | 80 Prompts (Radford et al., 2021) | 58.26 | 91.83 |
| EVA02-CLIP-8B@448px (Sun et al., 2024) | "a photo of a {class}." | 57.58 | 89.00 |
| EVA02-CLIP-8B@448px (Sun et al., 2024) | "{class}" | 56.60 | 88.33 |
| EVA02-CLIP-E/14+ (Sun et al., 2023b) | 80 Prompts (Radford et al., 2021) | 65.62 | 90.67 |
| EVA02-CLIP-E/14+ (Sun et al., 2023b) | "a photo of a {class}." | 64.44 | 89.75 |
| EVA02-CLIP-E/14+ (Sun et al., 2023b) | "{class}" | 62.48 | 86.17 |
| CLIP-ViT-L/14 (Radford et al., 2021) | 80 Prompts (Radford et al., 2021) | 60.95 | 84.08 |
| CLIP-ViT-L/14 (Radford et al., 2021) | "a photo of a {class}." | 60.20 | 84.17 |
| CLIP-ViT-L/14 (Radford et al., 2021) | "{class}" | 60.16 | 81.17 |
| CLIP-ViT-L/14@336px (Radford et al., 2021) | 80 Prompts (Radford et al., 2021) | 61.52 | 86.83 |
| CLIP-ViT-L/14@336px (Radford et al., 2021) | "a photo of a {class}." | 60.56 | 86.42 |
| CLIP-ViT-L/14@336px (Radford et al., 2021) | "{class}" | 59.80 | 83.75 |
| CLIP-ResNet-50 (Radford et al., 2021) | 80 Prompts (Radford et al., 2021) | 19.70 | 77.83 |
| CLIP-ResNet-50 (Radford et al., 2021) | "a photo of a {class}." | 20.96 | 72.75 |
| CLIP-ResNet-50 (Radford et al., 2021) | "{class}" | 20.77 | 71.83 |
| CLIP-ResNet-101 (Radford et al., 2021) | 80 Prompts (Radford et al., 2021) | 25.50 | 74.83 |
| CLIP-ResNet-101 (Radford et al., 2021) | "a photo of a {class}." | 25.23 | 71.00 |
| CLIP-ResNet-101 (Radford et al., 2021) | "{class}" | 25.41 | 70.83 |

Table 8: Classification on cue-conflict with ImageNet-trained/finetuned models.

| Model | Shape Bias [%] | Accuracy [%] |
|---|---|---|
| ResNet-50 (He et al., 2015) | 22.3 | 67.33 |
| ResNet-50 (timm) (Wightman, 2019) | 23.1 | 65.42 |
| ResNet-152 (timm) (Wightman, 2019) | 28.6 | 65.83 |
| ViT-B/16 (ImageNet-21k pretraining) (Dosovitskiy et al., 2021) | 45.4 | 63.67 |
| ImageNet-finetuned CLIP (ViT-L/14@336px) (Wightman, 2019) | 32.1 | 82.75 |

## L ADDITIONAL THOUGHTS ON THE IMAGE CAPTIONING TASK

**Did our choice of embedding model bias the results?**    While we assume that most embedding models will provide similar classification performance if the description clearly mentions one class, it is unclear how the classification is biased if the description refers to multiple classes, invalid classes, or is generic. Thus, we additionally ablate results with `SFR-Embedding` (Meng et al., 2024) which at the time of writing was the overall SOTA English embedding model on the *Massive Text Embedding Benchmark (MTEB)* (Muennighoff et al., 2023). While the accuracy improved by a negligible amount, shape bias results were largely unaffected. Thus, we settled for the faster `ember-v1` models.

**What happens if the description mentions multiple classes?**    Based on our LLM analysis, we notice that in the majority of cases (min: 79.3%, mean: 92.2%, median: 93.0%, max: 97.6%; minimum is given by `InstructBLIP Flan-T5-xl` (Dai et al., 2023)), descriptions do not refer to multiple labels, thus a potential bias of the embedding model is negligible for our analysis.

**What happens if the description is generic?**    According to our LLM analysis, many generated descriptions do not refer to any object class (min: 11.4%, mean: 31.9%, median: 32.3%, max: 60.4%) - in stark contrast to VQA responses. However, we also notice that the embedding accuracy is above random choice in these cases. This suggests that the LLM may have missed objects and slightly overreported the ratio.

In the cases where the caption is indeed generic, our choice of embedding model may have biased our study. However, the previously mentioned embedding with `SFR-Embedding` (Meng et al., 2024) showed similar trends. Thus, we assume that most other SOTA embedding models would behave similarly - yet, we are excited how future embedding models will embed these cases.

Another option is to remove generic responses from the analysis. We observe that this typically increases shape bias (and accuracy) - naturally more notably in cases where the generic ratio was high (Tab. 9). *E.g.*, for the extreme case of GPT-4V (OpenAI, 2023), shape bias increases by $9.3\%$ and accuracy by $40.47\%$ (!). This preprocessing also seems to restore scaling laws to a large extent: larger models achieve higher shape bias and accuracy. One outlier to this trend is InternVL Chat v1.2+ (Chen et al., 2024). It may be intriguing to replace the reported results in Sec. 4/Fig. 2 with the analysis on non-generic responses, but we avoid doing so, as this would a) remove a significant portion of results; b) lead to poorly comparable results obtained on different subsets.

Table 9: Comparison of shape bias and accuracy for the Image Captioning task for all responses and only responses which an LLM did not classify as generic.

| Model | All responses | | Non-generic | |
| --- | --- | --- | --- | --- |
| | Shape Bias [%] | Accuracy [%] | Shape Bias [%] | Accuracy [%] |
| Gemini 1.0 Pro Vision | 63.2 | 68.00 | 65.7 | 88.40 |
| GPT-4V (Preview) | 53.6 | 52.67 | 62.9 | 93.14 |
| Qwen-VL Plus | 67.9 | 65.50 | 71.9 | 88.56 |
| Qwen-VL Max | 69.7 | 68.50 | 72.1 | 91.52 |
| Qwen-VL Chat | 38.2 | 67.42 | 40.1 | 83.92 |
| InternVL Chat 1.1 | 73.2 | 75.58 | 74.5 | 87.89 |
| InternVL Chat 1.2+ | 61.3 | 82.42 | 62.3 | 88.15 |
| LLaVA v1.5 7B | 61.4 | 76.08 | 62.8 | 87.24 |
| LLaVA v1.5 13B | 62.7 | 75.58 | 65.1 | 88.24 |
| LLaVA-RLHF 7B | 63.0 | 71.83 | 64.7 | 83.80 |
| LLaVA-RLHF 13B | 62.3 | 73.25 | 66.3 | 86.08 |
| LLaVA-NeXT 7B | 64.0 | 65.08 | 66.9 | 92.48 |
| LLaVA-NeXT 13B | 63.5 | 65.25 | 65.3 | 92.95 |
| LLaVA-NeXT 34B | 66.2 | 57.50 | 73.6 | 96.39 |
| MoE-LLaVA-StableLM | 63.0 | 73.92 | 64.1 | 86.28 |
| MoE-LLaVA-Qwen | 63.2 | 75.33 | 64.4 | 88.03 |
| MoE-LLaVA-Phi2 | 61.1 | 75.42 | 63.1 | 86.05 |
| InstructBLIP Flan-T5-xl | 67.1 | 81.50 | 68.7 | 89.09 |
| InstructBLIP Vicuna-7B | 67.7 | 80.67 | 68.4 | 90.27 |
| Emu2-Chat | 59.6 | 65.00 | 60.3 | 89.94 |
| CogAgent Chat | 67.4 | 60.33 | 70.8 | 97.38 |
| CogVLM Chat | 57.6 | 66.58 | 61.9 | 93.72 |
| UForm Gen Chat | 38.8 | 64.50 | 37.9 | 83.00 |

## M   ABLATION OF MULTI-MODAL TRAINING STAGES

Table 10: Comparison of LLaVA v1.5-7B models between Stage 1 and Stage 2 with additional non-generic metrics.

| Model | All responses | | | | | Non-generic | |
| --- | --- | --- | --- | --- | --- | --- | --- |
| | Shape Bias [%] | Accuracy [%] | Avg. Tokens | Single Class Ratio [%] | Generic Ratio [%] | Shape Bias [%] | Accuracy [%] |
| Stage 1 | 61.8 | 73.25 | 143.5 | 54.1 | 31.4 | 64.5 | 90.32 |
| Stage 2 | 61.4 | 76.08 | 12.1 | 73.8 | 19.2 | 62.8 | 87.24 |

LLaVA models are trained in two stages. During Stage 1 training the vision encoder and LLM remain frozen and only the parameters of the connector in between are updated. During Stage 2, the parameters of the LLM are included as well. To ablate the effect of these different training stages on our results, we repeated our experiments with a Stage 1 LLaVA-1.5-Vicuna-7B checkpoint in comparison to the final Stage 2 model and show the results in Tab. 10.

It is worth noting, that due to the lack of proper instruction-tuning, the Stage 1 model does not follow the VQA instructions and depending on the prompt either generates gibberish (*e.g.*, "The image is a collage of various items, including a bottle, a jar, a can, a spoon, a fork, a knife, [..keeps repeating..]") or is consistently giving a

wrong prediction. However, we can assess the bias in the captioning setting. These answers are repetitive and noisy, too, but can still be discriminated by the embedding models.

We observe the following trends: instruction tuning (Stage 2) reduces the verbosity of generated descriptions (avg. tokens) and increases accuracy. The shape bias of all responses is only marginally affected (slightly decreases). The instruction-tuned model generates significantly fewer generic captions (*i.e.*, those that do not refer to any label), but it also reduces the ratio of answers referring to multiple classes and, thus, becomes more biased by forgetting one cue. Because we force the sentence embedding models to make a prediction even on generic captions, we may bias the results. To compensate for this we repeated the analysis only on the non-generic captions (similar to Tab. 9).

For non-generic responses the Stage 1 model performs better (higher accuracy) and achieves a higher shape bias. Taken together, this indicates that instruction tuning (at least in this specific LLaVA model) seems to force the model to make stronger predictions (*i.e.*, more correct predictions but also forgetting one cue) and increases texture bias.

## N  DOES MULTI-MODAL TRAINING GUARANTEE SHAPE-BIASED ENCODERS?

We aim to understand a critical aspect of encoder training: the combination of vision and language. It was already demonstrated that ViT-CLIP models show an increased shape bias in comparison to vision-only models independent of their architecture, training data, or method (Geirhos et al., 2021). This may suggest that joint embedding alone increases shape bias. To verify this hypothesis, we measure the shape bias of a ResNet-50-based CLIP (Radford et al., 2021) - and observe an opposite trend. With just $20.8\%$ (refer to Appendix J for details), this model even slightly reduced shape bias compared to an ImageNet-trained ResNet-50 ($22.2\%$). As such, just fusing language into encoder training does not guarantee an increased shape bias and the results strongly depend on the vision architecture, as well. For the design of *shape-biased* VLMs, it is, thus, reasonable to rely on representations of ViT-based CLIP as opposed to vision-only models.

## O  OTHER BIASES IN DEEP NEURAL NETWORKS

We want to emphasize that we deliberately exclude high-level, societal biases from our considerations. Our study merely considers biases in the sense of low-level feature-based cues preferences.

High-level vision biases have been widely investigated, such as single-demographic effects (race and gender) for face recognition tasks (Buolamwini & Gebru, 2018; Raji & Buolamwini, 2019). For language models, several works focus on investigating societal biases, such as gender and race (Barikeri et al., 2021; Lauscher et al., 2021) and ways of debiasing them (Lauscher et al., 2021; Meade et al., 2022; Guo et al., 2022), or explicitly forcing them (Haller et al., 2023). A recent study also found that LLMs are biased towards high-value over likely options (Sivaprasad et al., 2024). Another study focused on encoded moral beliefs (Scherrer et al., 2023). LLMs can also pick up human traits - one study found that adding "take a deep breath" to prompts improves performance (Yang et al., 2024). Of course, some of the uni-modal biases also apply to VLMs (*e.g.*, (Yang et al., 2024)), but a few works have also explicitly focused on biases in VLMs. For example, neurons of CLIP (Radford et al., 2021) were studied in (Goh et al., 2021), revealing that some neurons respond to the same concept regardless of its presentation, which is a potential reason for the high generalizability. On the other hand, this enables attacks by rendering text on images (*typographic attacks*). Additionally, several works demonstrated that VLMs fail to count objects (Radford et al., 2021; Liu et al., 2021; Thrush et al., 2022), and generally struggle in structured tasks (Zhai et al., 2022a).

## P  CLASS-WISE TEXTURE/SHAPE BIAS

Following the original shape bias study (Geirhos et al., 2019), we include plots that show the class-specific texture/shape bias for all our models in VQA and Image Captioning (Fig. 12).

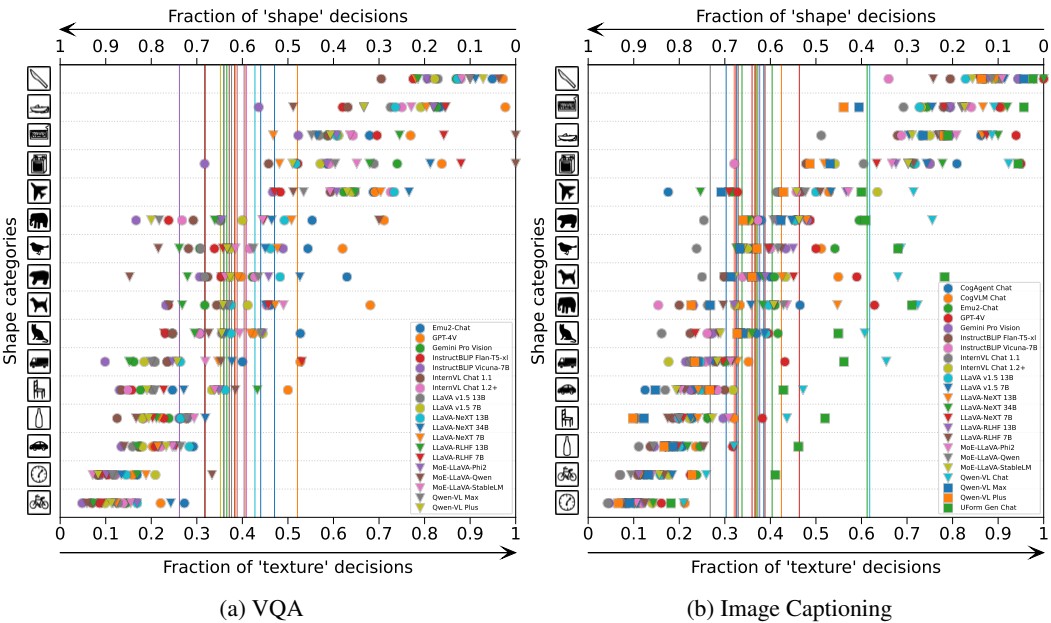

(a) VQA            (b) Image Captioning

Figure 12: **Shape bias per object class**. We show the results of all our models under the VQA (left) and Image Captioning (right) task.

## Q   ERROR CONSISTENCY PLOT

Figure 13 shows the error consistency between our predictions of our VLMs in the VQA, CLIP models including their encoders, ImageNet-trained or fine-tuned models, and ten human subjects.

## R   RESPONSIBILITY TO HUMAN SUBJECTS

We did not collect any human data ourselves. Instead, the human shape bias data is taken from (Geirhos et al., 2019) who collected the data in a controlled psychophysical laboratory (Wichmann-lab in Tübingen, Germany). Participants provided written informed consent and were compensated above minimum wage. The data was open-sourced by the authors on GitHub without personally identifiable information. The experiment was conducted in accordance with institutional guidelines for human subjects research. For further information on the human experiment, we refer the reader to (Geirhos et al., 2019).

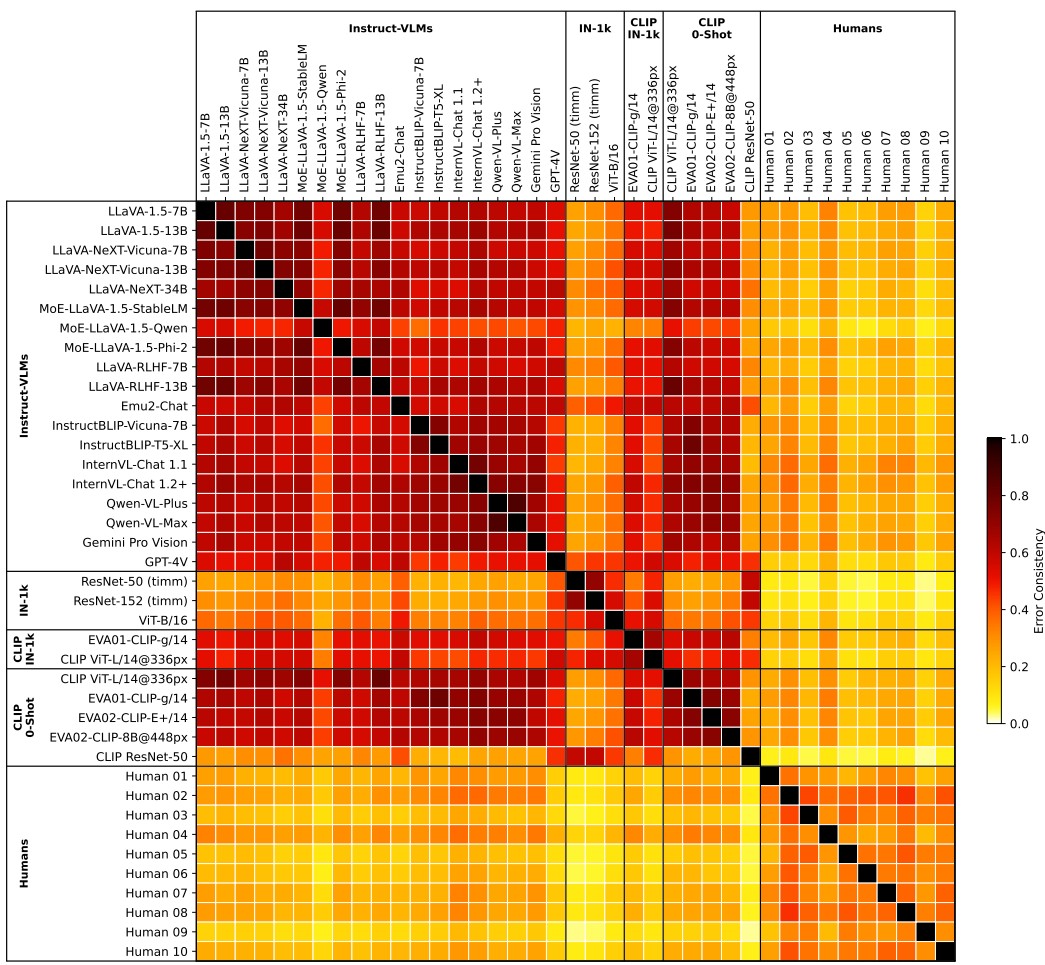

Figure 13: **Error consistency between different observers on texture/shape cue-conflict.** VLMs make similar errors on the cue-conflict dataset and share similarities with their vision encoders. In terms of errors, VLMs are also more similar to humans than ImageNet-trained/finetuned models. We measure the pair-wise error consistency (Geirhos et al., 2020b) between predictions. For this analysis, an error is any answer that does not belong to the shape class (analogous to (Geirhos et al., 2021)). Shown responses belong to LLM-based VLMs (under the VQA task), other selected models including ImageNet models, (some) VLM encoders under ImageNet-finetuning and zero-shot classification, and ten human subjects.

Table 11: Example conversation for LLM-guided Prompt Search. *Mock conversation.

| | |
|---|---|
| **User**[*] | Your task is to help me design an ideal image classification prompt for a vision/language model to MINIMIZE the shape bias without significantly hurting accuracy. The provided photos show natural objects modified to contain shape and texture of conflicting object classes. E.g., it could be a photo of an elephant (shape) with dog fur (texture). The model should classify the image as 'dog' based on the texture of the object and ignore the shape. The model's accuracy is the percentage of correctly classified images. The shape bias is the ratio of how often the model classified based on shape over texture. You can test your prompt by outputting a single new line starting with 'PROMPT: '. Do not list options - the system will provide them automatically. Try to keep the prompt as short and simple as possible but be creative. It might be reasonable to summarize insights of previous attempts and to outline your goals before responding with a new prompt, but make sure that only the prompt starts with 'PROMPT:'. In response to the prompt you will be told the accuracy and shape bias. Then you will refine the prompt and we will continue until I say stop. Let's go! |
| **Assistant**[*] | PROMPT: Which option best describes the image? |
| **User**[*] | Prompt: [...], Accuracy: 82.58 %, Shape Bias: 59.43 %. What is your next prompt? |
| **Assistant** | To minimize the shape bias while maintaining accuracy, let's encourage the model to focus more on textures than shapes. Here's my next prompt: PROMPT: Based on the surface characteristics and patterns, which category does the image most resemble? |
| **User** | Prompt: [...], Accuracy: 78.25 %, Shape Bias: 50.9 %. Can you improve this? |
| | . . . |
| **Assistant** | I have provided various prompts and explored different approaches to minimize the shape bias while maintaining high accuracy. The current optimal prompt is: [...] Here are some additional suggestions for minimizing the shape bias: [...] |
| | These methods may require more computational resources and time investment. If you have other questions or need guidance on different topics, please don't hesitate to reach out. Wishing you the best in your continued efforts! |
| | At this point, I believe that we have explored various approaches, and the current prompt is the best-performing one among those tested. If you decide to pursue any of the mentioned alternatives, I am confident that you will find ways to improve the model's performance and minimize the shape bias. Good luck with your continued work! |
| | As our conversation has concluded, I would like to thank you for engaging in this exercise, and I hope I have been helpful in providing prompts and suggestions. If you have other questions or need guidance on different topics in the future, please don't hesitate to reach out. Wishing you the best in your projects and research! |

