# OpenReview forum: "Can We Talk Models Into Seeing the World Differently?"
_ICLR.cc/2025/Conference — ICLR 2025 Poster_

### Official Review · Reviewer_1KNs · 2024-10-28

**Soundness:** 3
**Presentation:** 3
**Contribution:** 2
**Rating:** 6
**Confidence:** 4

**Summary:**

This paper investigates the shape-texture biases in recent vision-language models (VLMs). The author discovered that training with LLMs and vision-language tasks change the shape-texture biases, using the tasks of VQA and image captioning. However, the shape-texture bias is still influenced by the upstream vision-only encoders (e.g., CLIP). Based on such observations, the authors proposed both manual prompts and LLM-generated prompts to elicit the models' biases toward either shape or texture without significantly degrading the accuracy.

**Strengths:**

* S1: Overall, I think the paper is written clearly and highlights several key contributions.

* S2: The authors have conducted experiments on a wide range of VLMs and reach the interesting conclusion that -- VLMs influences the shape texture biases compared with their upstream vision-only encoders. I think the analysis is thorough and detailed in this part, covering multiple scenarios (VQA and image captioning), diverse models, and several comparative experiments (e.g., in Table 1). I am not an expert in explainability research, but this paper is the first to study VLMs' shape-texture biases to the best of my knowledge.

* S3: The authors demonstrate the stability of VLMs' biases towards either shape or texture (Figure 3) and how to steer their behaviors (Figure 4 and Figure 5). Both manual prompt tuning and automatic LLM optimization can achieve such steering. According to the authors' claim, this might be potentially useful.

* S4: The analysis and experiments in this paper are detailed and comprehensive.

**Weaknesses:**

* W1: After reading the paper, it is still unclear to me **what the significance of studying shape and texture biases is**, primarily when MLLMs can already address more complicated tasks than object recognition nowadays. This concern limits the significance of this paper. Perhaps this point sounds biased, but I wish the authors had provided more arguments or examples of how studying shape-texture biases benefits the development or applications of VLMs. Such contexts would help the readers not to focus specifically on shape-texture biases.

* W2: The conclusions mentioned in the paper did not seem explicit from the quantitative results. For example, Table 1 does not show conclusive observations. If the goal of this table is to show that VLMs can behave very differently than their original visual encoder, this does not seem to be a surprising discovery since VLMs are trained on different datasets with different recipes compared with the original vision encoder.

* W3: More clarifications are needed on the steering part, where I have similar concerns with W1 -- the authors might want to clarify the usefulness of the potential contribution of steering such biases. (1) The claim in L448-449, "overall accuracy does not change considerably," might indicate that shape-texture bias is not a significant factor for accurate models. (2) With the recent progress in prompt engineering, the authors might want to explain better "why prompts change the model behaviors," which is non-trivial.

* W4: With this paper focusing on analysis, a crucial yet missing analysis is to rule out the biases from LLMs. For example, if the prompt mentioned in the paper contains the words of "textures" or "shapes," will the shape/texture biases observed related to the prior knowledge in the LLMs. It would strengthen the paper if authors can assist the analysis with some attention maps or other preferred measure to show that the **visual representation indeed changes** under such shape/texture steering.

**Questions:**

Please see above for questions. The importance would be W1 --> W3 --> W4 --> W2. Overall, better clarifying the significance of this study would improve my scoring.

---

> ### Author Response · Authors · 2024-11-18
> **Rebuttal (1/2)**
>
> Dear Reviewer 1KNs,
> We thank the reviewer for the review and valuable feedback. We appreciate the assessment of our work as novel due to **”being the first to study VLM’s shape-texture biases”**, with a “**detailed and comprehensive analysis**” showing **”several key contributions**” in a **”clearly written”** paper. Please find detailed answers to each comment below.
>
> > W1: After reading the paper, it is still unclear to me what the significance of studying shape and texture biases is, primarily when MLLMs can already address more complicated tasks than object recognition nowadays. This concern limits the significance of this paper. [...] how studying shape-texture biases benefits the development or applications of VLMs.
>
> We apologize that our submission did not communicate this more clearly, and would like to break down this comment into multiple parts for clarity.
>
> **Goal of the paper:**
> First, we would like to clarify that the goal of this paper is to show that purely visual biases can be steered by language (at inference time). This is a powerful and yet very cheap method to de-bias a model without having to retrain it. We use the texture/shape bias as one specific instance, especially since it is potentially the best-researched vision bias. However, the emphasis is not on that bias alone.
>
> **Why is texture/shape significant:** The texture/shape bias fundamentally differs in how models and humans process vision. Human perception relies heavily on shape to identify objects, while many models typically show a strong texture bias, relying more on local patterns rather than global shapes. This difference can lead to models performing well on certain (i.i.d.) datasets but failing to generalize in real-world scenarios where shapes are more consistent than textures. By adjusting models to have a (stronger) shape bias, they align more closely with human vision, and we expect them to better generalize and be more interpretable due to the increased similarity.
>
> **VLMs already address more complicated tasks:** VLMs support many advanced tasks, such as Visual Question Answering (VQA). However, these tasks often rely fundamentally on object detection, meaning that any bias in the object detection process will transfer to these tasks. Studying a bias in such a low-level task can be a fruitful way to improve performance in many more abstract tasks that are dependent on object recognition.
>
> **How do VLM development/applications benefit:** Our priority was to show that the model can be debiased (to some extent) at inference time by just using language. This is very much in line with the zeitgeist to optimize large foundation models at inference time (e.g., model editing [1] or test-time scaling [2] as done in GPT-o1) and not having to retrain them from scratch e.g., to remove biases.
>
> However, our study also carries some takeaways for development:
> Whenever possible a debiased vision encoder should be used as these are more steerable. This is of course easier said than done, but being aware of specific biases through a large-scale study like ours is a step in the right direction.
> We have seen that LLMs turn flexible vision representations (i.e., ones that contain both cues) into highly biased predictions by completely ignoring one cue. This requires a lot of attention in future training - LLMs should clearly not just ignore some features.
>
> Does this clarify the significance, and would the reviewer recommend that we add (parts of) our explanation to the paper?
>
> > W3: More clarifications are needed on the steering part, where I have similar concerns with W1 -- the authors might want to clarify the usefulness of the potential contribution of steering such biases. (1) The claim in L448-449, "overall accuracy does not change considerably," might indicate that shape-texture bias is not a significant factor for accurate models.
>
> We hope to have clarified the usefulness above and want to focus on what performance means in our study, as there seem to have been some misunderstandings.
>
> The “performance” we report is the accuracy measured on the texture/shape cue-conflict dataset. Predicting either the shape **or** texture label counts as a correct prediction. This serves as an important signal for our study, showing that our prompting strategies indeed *steer* the bias in outputs (* see below), but it does not tell us if this “performance” transfers to other problems. Other works have shown that the texture/shape bias is, for example, strongly correlated with ImageNet accuracy [3].
>
> *  For example, you could get perfect shape bias by just predicting a false label for all detections that are currently texture-biased – however, this is not steering as it comes at a significant cost in accuracy. We clarified this in L149ff, as our current phrasing seems to have confused multiple reviewers.

---

> > ### Author Response · Authors · 2024-11-18
> > **Rebuttal (2/2)**
> >
> > > (2) With the recent progress in prompt engineering, the authors might want to explain better "why prompts change the model behaviors," which is non-trivial.
> >
> > Could the reviewer kindly clarify what they specifically refer to by “recent progress in prompt engineering”?
> >
> > Conceptually, multi-modal training aims to learn a correlation between vision and text tokens, that the LLM leverages to predict the output token stream. Our steering experiments establish that the investigated VLMs have learned a correlation between shape and texture vision and text tokens, and not only a superficial but quite a semantically robust one, as can be seen through the experiments on synonyms.
> >
> > This differs from traditional prompt engineering in two ways: we show that (1) this multi-modal correlation can recalibrate representation biases of the vision encoder - which is a novel finding to the best of our knowledge and (2) the correlation not only holds for high-level representations (like entire objects) but also applies to low-level features like texture and shape cues.
> >
> > > W4: With this paper focusing on analysis, a crucial yet missing analysis is to rule out the biases from LLMs. For example, if the prompt mentioned in the paper contains the words of "textures" or "shapes," will the shape/texture biases observed related to the prior knowledge in the LLMs.
> >
> > Thank you very much for this interesting pointer! It is worth noting that the visual representation is generated by a vision encoder that is independent of the prompt. Inside the LLM, we can only access an entangled representation of all modalities which is hard (if at all) interpretable – that remains an open research question.
> >
> > However, we think that our experimental results in Fig. 4 actually answer your question: If we specifically bias the prompt we get an equivalently biased response. We also ensured that this (global) representation is not superficial by using synonyms: synonyms of texture are more texture-biased and synonyms of shape are more shape-biased. Does this address your question?
> >
> >
> > > W2: The conclusions mentioned in the paper did not seem explicit from the quantitative results. For example, Table 1 does not show conclusive observations. If the goal of this table is to show that VLMs can behave very differently than their original visual encoder, this does not seem to be a surprising discovery since VLMs are trained on different datasets with different recipes compared with the original vision encoder.
> >
> > The goal of this particular table was indeed to show that the VLM behaves differently than the encoder, but ultimately, it is just a mechanistic analysis to motivate why steerability works.
> >
> > Now whether or not a finding is “surprising”, of course, heavily depends on one’s prior expectations and while we cannot change a reader’s prior expectations, we certainly can provide evidence to support a “posterior” assessment, supported by data as opposed to expectations. Of course, a change in accuracy is not surprising - what is surprising is the **change of the bias** – especially given that all the encoders in that table were **not fine-tuned** during multi-modal training. Overall, our research demonstrates that language, despite being conceptually unrelated to visual biases like texture/shape bias, can significantly influence the bias in the final prediction of a VLM. This interaction between language and visual processing in VLMs is a novel finding and, as far as we are aware, not one that directly follows from any existing work.
> >
> >
> > [1] Cao et al., “Editing Factual Knowledge in Language Models”, EMNLP, 2021.
> >
> > [2] Snell et al., “Scaling LLM Test-Time Compute Optimally can be More Effective than Scaling Model Parameters”, Preprint, 2024.
> >
> > [3] Gavrikov et al., “Can Biases in ImageNet Models Explain Generalization?”, CVPR, 2024.
> >
> > Once again we thank you for your valuable feedback! We look forward to further discuss this if there are remaining questions and thank you for your careful consideration of our paper.

---

> > > ### Comment · Reviewer_1KNs · 2024-11-22
> > >
> > > I think my concerns are resolved after reading the author's rebuttal and the other reviewers' opinions. I will raise my scores.

---

### Official Review · Reviewer_Ume1 · 2024-11-03

**Soundness:** 4
**Presentation:** 4
**Contribution:** 3
**Rating:** 6
**Confidence:** 3

**Summary:**

This paper shows that VLMs still preserve the texture and shape biases of image encoders, however, VLMs are more biased in shape rather than texture.
It also shows the results compared to the image encoder including ReSNet-50, and CLIP.
Also, this paper presents the methodology for controlling those biases through steering prompts.

**Strengths:**

- This paper is well-written and well-organized.
- Extensive study to study textual and shape biases in VLMs.
- Also, this paper presents a steering prompt that may change the bias.
- This paper supplements detailed analysis and implementation details.

**Weaknesses:**

- In section 5.2, spectral biases or critical bands that are difficult to control in language, I ask for more of the authors' opinions on why the language prompt influenced this bias and what it means when the performance decreases even if the bias value changes.

- According to the results of Figure 4, steering can slightly control textual/shape biases, but it seems to have a slight impact on overall performance, so what is the point of controlling these biases?

**Questions:**

- What is the base performance of the original image encoder in Table 2?

---

> ### Author Response · Authors · 2024-11-18
> **Rebuttal (1/2)**
>
> Dear Reviewer  Ume1,
> We would like to thank the reviewer for the review and valuable feedback. We are happy to hear that the reviewer found the article **”presented excellent”**, providing an **”extensive study”** with a **”detailed analysis in the supplementary”**. Please find detailed answers to each comment below.
>
>
> > W1: In section 5.2, spectral biases or critical bands that are difficult to control in language, I ask for more of the authors' opinions on why the language prompt influenced this bias
> In Sec. 5.2, we showed that steering the low-/high-frequency bias is possible, but not as effective as the texture/shape-bias.
>
> If the question is “why does this work at all”: we want to clarify that we do not think that spectral biases are not controllable at all, it is just that there is likely less multi-modal training available for this bias. After all, our results show **statistically significant changes**.
>
> Conceptually, for language prompt steering to work properly, we need to satisfy (at least) two conditions: (1) the vision encoder must provide a “flexible” representation, which allows reconstruction of both feature cues, (2) the LLM must have learned a correlation between natural language and the visions tokens. Based on the vision encoder bias (see answer to Q1), we can assume that the representation is flexible and the problem must be more related to condition (2). This is, intuitively, reasonable, as learning texture or shape features is directly possible from pixel values, but understanding frequency bands requires a Fourier transform. Additionally, spectral effects are much harder to describe via natural language and probably underrepresented in the text training data. So, it is less likely that the VLM saw text/image pairs connecting these features.
>
> Does this answer your question?
>
> > and what it means when the performance decreases even if the bias value changes.
>
> The simple answer is that the steering is a bit noisy (like many things in VLMs).
>
> Recall that we have two labels for each cue-conflict sample, e.g., one for shape and texture. Accuracy is unusually defined by the ratio of predictions where the model has predicted *either one* of the labels (i.e., made no error). When we “steer” the bias, we would like the predictions to change but not for the model to make additional errors. For example to steer toward shape bias, ideally, the ratio of shape predictions would increase at the same rate as the ratio of texture predictions would drop. This would show that the model has indeed shifted its bias and is prioritizing features differently now. In that case, the accuracy would not change – but most systems are noisy in practice and it seems that occasionally we also destroy the attention to some features.
>
>
> > W2: According to the results of Figure 4, steering can slightly control textual/shape biases, but it seems to have a slight impact on overall performance, so what is the point of controlling these biases?
>
> The “performance” we report is the accuracy measured on the texture/shape cue-conflict dataset. Predicting either the shape **or** texture label counts as a correct prediction. This serves as an important signal for our study, showing that our prompting strategies indeed *steer* the bias and not just ignore textures in outputs (see answer above) – however, it does not tell us if this performance difference (if any) transfers to other problems. Specifically, on ImageNet samples we would not expect any improvements in accuracy from a shape bias, as the task is solvable by texture alone [1]. However, in any task where the bias is uncorrelated with the true label at test time, you should see improvements.
>
> To give a bit more perspective on why we want to control biases:
> Recently, many biases from low-level vision (like texture/shape) to high-level societal biases (gender bias) were discovered in SOTA models (see Appendix O). Effectively, a bias means that a model is over-prioritizing some feature cue. Some biases are benign, for example in ImageNet classification - where we often have one single object in front of a background - a hypothetical “foreground bias” is exactly what we want to detect objects, but on the other hand, a “background bias” would be unwanted. These biases often arise due to “shortcuts” in training data [2] and even a biased model may perform well on i.i.d. datasets. However, when we sample more data at test time when the shortcut is no longer present, it would cause a drop in model performance.
>
> In this paper, we show that language (via prompts) can be a simple tool to steer unwanted biases toward expected behavior and actual generalization. This method is cheaper than (re)training a de-biased model, especially for expensive training as done in VLMs.
>
> [1] Brendel et al., “Approximating CNNs with Bag-of-local-Features models works surprisingly well on ImageNet”, ICLR, 2019.
>
> [2] Geirhos et al., “Shortcut Learning in Deep Neural Networks”, Nat. Mach. Int., 2020.

---

> > ### Author Response · Authors · 2024-11-18
> > **Rebuttal (2/2)**
> >
> > > Q1: What is the base performance of the original image encoder in Table 2?
> >
> > Thank you for the pointer! We are happy to report the LLaVA encoder's performance on frequency-cue-conflict and add this to the paper. Unfortunately, we cannot provide the results for InterVL 1.1 as there is no compatible text encoder available for that model, and additionally, the vision encoder is fine-tuned during training.
> >
> > Results:
> > The difference between encoder and VLM is even more pronounced for this bias and model – there is a 9.3% difference in low-frequency bias between the neutrally prompted VLM and encoder outputs. This once again illustrates that the LLM plays a role in steering the visual bias it inherited from the encoder.
> >
> > | VLM                | Encoder                       | LF-Bias | Accuracy |
> > |---------------------|-------------------------------|---------|----------|
> > | LLaVA-NeXT-7B      | CLIP VIT-L/14@336px (frozen)  | 61.7  %  | 86.8 %    |
> >
> > Once again we thank you for your valuable feedback! We look forward to further discuss this if there are remaining questions and thank you for your careful consideration of our paper.

---

> ### Comment · Reviewer_Ume1 · 2024-11-20
> **Thank you for authors' feedback!**
>
> I have understood what the authors struggle to study in section 5.2, and I have no other questions.

---

### Official Review · Reviewer_bGZu · 2024-11-04

**Soundness:** 3
**Presentation:** 3
**Contribution:** 3
**Rating:** 8
**Confidence:** 4

**Summary:**

- The paper investigates how visual biases in vision-language models (VLMs), specifically the shape vs. texture bias, are affected by integrating a large language model (LLM) with a vision encoder.
- It highlights that VLMs inherently inherit some biases from their vision encoders but exhibit unique behaviors when processing visual cues, influenced by the multi-modal fusion of vision and language.
- The study shows that VLMs tend to favor shape-based recognition over texture, unlike traditional vision-only models, but do not match the high shape bias seen in human perception.
- The authors demonstrate that these biases can be steered using natural language prompts without retraining, allowing adjustments in model behavior to prioritize shape or texture-based decisions.
- The research provides insight into the potential of language prompting to align model outputs with user preferences, showcasing a new, efficient method to adjust visual biases in multi-modal models.

**Strengths:**

- Introduces a new investigation into steering visual biases in VLMs using natural language prompts, extending beyond traditional uni-modal bias studies.
- Comprehensive experiments using the texture/shape cue-conflict dataset validate findings and provide robust evidence for claims.
- Clear organization with effective figures and tables to illustrate results, making complex concepts more understandable.
- Demonstrates that visual biases can be influenced through prompts without retraining, offering practical implications for user-aligned AI outputs.
- Opens pathways for further research on user-driven customization and bias control in multi-modal AI, enhancing model adaptability and transparency.

**Weaknesses:**

- Focus on shape and texture biases limits the study; exploring other biases (e.g., color, spatial attention) would give a fuller understanding of VLM behavior.
- Limited discussion on prompt selection; more detail on prompt strategies and robustness would enhance the study's depth.
- Prompt-based bias steering is constrained; deeper analysis on why some models are less responsive and how to improve steerability is needed.

**Questions:**

- Have you considered extending the study to include other visual biases such as color or spatial attention for a broader understanding of VLM behavior?
- Can you share additional examples or analyses of failure cases where the model showed unexpected behavior or limited steering?

---

> ### Author Response · Authors · 2024-11-18
> **Rebuttal (1/2)**
>
> Dear Reviewer bGZu,
> We would like to thank you for your kind review and your valuable feedback inspiring future analysis.  We appreciate your assessment of our work as **“opening pathways for further research”**, **“extends beyond traditional uni-modal bias studies”** and allows for **"more understanding of complex concepts”**, which  **“offers practical implications”**. Please find detailed answers to your comments below.
>
> > W1: Focus on shape and texture biases limits the study; exploring other biases (e.g., color, spatial attention) would give a fuller understanding of VLM behavior.
> > Q1: Have you considered extending the study to include other visual biases such as color or spatial attention for a broader understanding of VLM behavior?
>
>
> We totally agree, there is an exciting array of other biases to explore! In this paper, our primary emphasis was on *steerability* of VLM biases at inference time by language. Given our finite budget, we chose to prioritize more models instead of biases as there is always a risk that a bias is model-specific and limited our analysis to shape/texture and the transferability to low/high-frequency biases (Sec 5.2).
>
> However, we indeed do have plans to provide a more fundamental analysis of low-level vision biases in future work, for VLMs and beyond (independent of steering). As a little teaser, we show the following biases on LLaVA-NeXt-7B, based on the cue-conflict stimuli methodology:
>
> | Bias                              | LLaVA-NeXT-7B                  |
> |-----------------------------------|--------------------------------|
> | Color (red vs. green vs. blue)    | 30.12%, 60.62%, 9.26%         |
> | Color (hue vs. saturation vs. lightness) | 1.42%, 0.85%, 97.73%       |
> | Light Intensity (dark vs. bright pixels) | 50.84%, 49.16%             |
> | Spectral (amplitude vs. phase)    | 98.45%, 1.55%                 |
> | Spatial (top half vs. bottom half)| 42.48%, 57.52%                |
> | Spatial (left half vs. right half)| 51.38%, 48.62%                |
>
>
> Most of the findings align well with our expectations (i.e., no significant bias for left, right, or no information from H/S channels in the HSL space). It is quite impressive that LLaVA almost perfectly models human color perception (cf. ITU-R BT 601). One other bias we find is that LLaVA over prioritizes the lower half of an image by almost 7.5%.
>
> We are curious for your feedback and thank you again for this excellent pointer!
>
>
> > W2: Limited discussion on prompt selection; more detail on prompt strategies and robustness would enhance the study's depth.
>
> We would kindly point the reviewer to Appendix B, where we have already explored alternative prompts for VQA and Captioning. Additionally, we also show the accuracy/bias tradeoff of LLM-generated prompts in Appendix Fig. 6, which may answer the question about “robustness”. But of course, the space of possible prompts is infinite. Is there something specific that the reviewer would like us to try?
>
> > W3: Prompt-based bias steering is constrained; deeper analysis on why some models are less responsive and how to improve steerability is needed.
>
> Another great suggestion!
>
> For language prompt steering to work properly, we need to satisfy (at least) two conditions: (1) the vision encoder must provide a “flexible” representation, which allows reconstruction of any feature cues, (2) the LLM must have learned a correlation between natural language and the visions tokens.
>
> Evidently, these conditions may only be partially met in some models depending on the data and training method. For example, if the multi-modal training data does not contain any correlations between texture/shape and corresponding vision tokens, we would not expect steering to work at all.
>
> Based on these two conditions, we can formulate two methods to improve steerability:
> Use a (as much as possible) debiased vision encoder. The simplest (but often unrealistic) option to achieve this is to scale up the training data.
> Include multi-modal training data that correlates language and vision for the bias under test.
>
> Would the reviewer recommend adding this explanation to the paper?

---

> > ### Author Response · Authors · 2024-11-18
> > **Rebuttal (2/2)**
> >
> > > Q2: Can you share additional examples or analyses of failure cases where the model showed unexpected behavior or limited steering?
> >
> > Sure!
> >
> > In our automated prompt search, we have seen many generated prompts that did not or oppositely steer the bias. This is not very surprising, as language models are very brittle with respect to prompts.
> >
> > To give an example of conflicting steering, “Choose the correct category according to the predominant shape, even if there are misleading textures or patterns involved.” actually increases texture bias on LLaVA-NeXT-7B slightly (by ca. 4%). This is relatively rare and not necessarily a problem when using automated prompt search to discover steerable prompts, but may discourage human prompt engineering if the users overly anthropomorphize the models’ representation space.
> >
> > Other prompts were just unreasonably bad. For instance, the automated prompt search tried to add verbal examples, which sounds like a good idea given the promising results of in-context learning: *“In this exercise, prioritize the prominent textures in the images to ensure accurate classifications of the natural objects. Although shapes and forms are informative, place greater emphasis on the prevailing texture when deciding the categories. For instance, if an image demonstrates an elephant-like figure predominantly shrouded by dog-fur texture, label it as 'dog', emphasizing the significance of texture over shape in evaluating the shape bias.”* However, on LLaVA-NeXT-7B this reduced the accuracy to random chance (15%). This aligns with a very recent paper that showed that in-context learning by visual examples is not effective in steering shape bias [1].
> >
> > Beyond that, we generally found small Qwen-LM-based models to operate very poorly on the texture/shape cue-conflict dataset, although they seem to operate just fine in common benchmarks. For instance, comparing the results for MoE-LLaVA with different LLMs (see Tab. 4 in the Appendix), the Qwen-LM-based model is over 20% less accurate. Upon inspection of the predictions, we found that the Qwen model generally follows the prompt's instruction but has a much stronger tendency to hallucinate - specifically, it hallucinates the label “bird” very often, but never or rarely predicts “keyboard”, “knife”, or “oven”.
> >
> > We would like to point out that we provide all the raw responses and prompts in the supplementary materials!
> >
> > [1] Hemmat et al., “Hidden in Plain Sight: Evaluating Abstract Shape Recognition in Vision-Language Models”, NeurIPS, 2024.
> >
> > Once again we thank you for your valuable feedback! We look forward to further discuss this if there are remaining questions and thank you for your careful consideration of our paper.

---

### Official Review · Reviewer_VmoU · 2024-11-06

**Soundness:** 4
**Presentation:** 4
**Contribution:** 4
**Rating:** 8
**Confidence:** 4

**Summary:**

This paper studies vision-only biases in texture vs. shape and the preference for local over global information in recent vision-language models (VLMs). The paper reveals that VLMs partially preserve the shape bias from their vision encoders, though they do not reach human levels of shape reliance. The vision encoder produces a biased representation containing both texture and shape cues, which the LLM often suppresses, leading to recognition based on either shape or texture alone. At the last part, this paper shows that VLMs enable steering of visual biases through language prompt, allowing shape bias to be adjusted significantly.

**Strengths:**

- The paper is well-written with extensive analysis.

- The motivation is clear, and the problem is well-defined. It’s a pleasure to read.

- A comprehensive study is included, with inspiring and interesting findings.

**Weaknesses:**

- Class-wise analysis: For VQA classification, where tasks involve multiple classes, a class-wise analysis to examine texture and shape bias for each class would be helpful. Questions to explore include whether this texture and shape bias is consistent across all classes and whether there are differences between classes with simple vs. complex shapes.

- Additional qualitative results and examples: This paper provides quantitative results across various aspects; however, it lacks qualitative results and examples, which would help clarify the concepts through visual examples. For instance, the reviewer suggests including examples from the target dataset with texture and shape splits. Additionally, presenting qualitative examples of both correct and incorrect predictions could further enhance the paper’s clarity and impact.

**Questions:**

- The results of steering the texture/shape bias via language prompts are interesting. The reviewer observes that steering towards a texture bias can lead to completely different model behaviors for different models. For example, the performance of Gemini increases by 1%, while LLaVA-NeXT-7B decreases by 2%. However, in the study of prioritizing shapes over texture cues in VLMs, LLaVA-NeXT-7B actually shows a stronger texture bias than Gemini. Do the authors have any comments or findings in this direction?

---

> ### Author Response · Authors · 2024-11-18
> **Rebuttal**
>
> Dear Reviewer VmoU,
> We thank the reviewer for the review and the valuable feedback. We are happy to hear that the reviewer found the article a **"pleasure to read"** with **"inspiring findings"**, a **"clearly provided motivation"** and of  **"excellent soundness"**. Please find a detailed answer to the comments below.
>
> > W1: Class-wise analysis: For VQA classification, where tasks involve multiple classes, a class-wise analysis to examine texture and shape bias for each class would be helpful. Questions to explore include whether this texture and shape bias is consistent across all classes and whether there are differences between classes with simple vs. complex shapes.
>
> We kindly point the reviewer to our class-wise analysis in Appendix O - Figure 11 for both VQA and Image Captioning. The original Texture/Shape-Bias paper (Geirhos et al., 2019)  already observed that there are class-wise differences for discriminative classifiers, and this continues to hold for VLMs. For instance, you can see that the “bicycle”- class is highly shape-biased, while “knife” tends to be mostly texture-biased. For some classes, like “elephant” or “oven” there is quite some variance depending on the model. Since we are dealing with natural objects, it is unclear to us how we can faithfully identify if objects have simple or complex shapes. But our subjective analysis is, if we look at the top-3 most shape-biased classes (“car”, “bicycle”, “clock”) we see objects with highly **distinct** shapes but different levels of complexity. This may suggest that distinctness is more important than complexity for shape bias.
>
> Has the reviewer any specific suggestion in mind, or does this already answer the question?
>
> > W2: Additional qualitative results and examples: This paper provides quantitative results across various aspects; however, it lacks qualitative results and examples, which would help clarify the concepts through visual examples. For instance, the reviewer suggests including examples from the target dataset with texture and shape splits.
>
>
> Thank you for your suggestion! We have now extended Fig. 9 in the appendix to show a few examples from the texture/shape cue-conflict dataset, in addition to the samples from frequency-cue-conflict. Does this address your comment?
>
>
> > Additionally, presenting qualitative examples of both correct and incorrect predictions could further enhance the paper’s clarity and impact.
>
> We appreciate another great suggestion! It is a bit difficult to do this consistently, as we show many models, but we identified “agreement sets”, where all VLM predict shape, texture, correct or wrong prediction, respectively. We have added this analysis in a new Appendix chapter: I “Agreement sets on texture/shape cue-conflict”,
>
>
> > Q1: The results of steering the texture/shape bias via language prompts are interesting. The reviewer observes that steering towards a texture bias can lead to completely different model behaviors for different models. For example, the performance of Gemini increases by 1%, while LLaVA-NeXT-7B decreases by 2%. However, in the study of prioritizing shapes over texture cues in VLMs, LLaVA-NeXT-7B actually shows a stronger texture bias than Gemini. Do the authors have any comments or findings in this direction?
>
> Thank you for calling our results interesting!
>
> It is worth clarifying that the “performance” we show is the accuracy on the *texture/shape cue-conflict* dataset – i.e., we count a prediction as correct if it matches either the shape or texture label. If we would perfectly steer the model, we would see label flips where texture decisions become shape or vice versa. In that case, the accuracy would be the same.
>
> If the accuracy decreases it means that we have not evenly steered the bias - given the few samples, 1-2% are however not significant. If the accuracy increases, it means that we now classify samples correctly that we previously hadn’t – this can happen if one feature cue is spuriously correlated within the decision rule of the model. We do not know what encoder Gemini uses, but we would assume that it is different from LLaVAs encoder, and is likely trained on different data, which results in different correlations between features and labels that reflect this different behavior.
>
> Once again we thank you for your valuable feedback! We look forward to further discuss this if there are remaining questions and thank you for your careful consideration of our paper.

---

> > ### Comment · Reviewer_VmoU · 2024-11-27
> >
> > Thanks for the response from the authors. My concerns have been addressed in the author's rebuttal, and I will increase my scores.

---

### Author Response · Authors · 2024-11-18
**Global Response**

We thank all reviewers for their valuable feedback. We are pleased to hear that all reviewers enjoyed our paper calling it **”a pleasure to read/inspiring and interesting findings”** (VmoU), **“comprehensive experiments/clear organization”** (bGZu), **“well-written and well-organized/extensive study/detailed analysis”** (Ume1), and **“analysis and experiments [..] are detailed and comprehensive.“”** (1KNs). Based on the feedback, we have implemented some changes in the PDF to solidify our study (highlighted in bright green). While we discuss these in detail in the individual comments, we want to give a brief overview here:

1. Clarified the meaning and implication of changes in the “cue accuracy”  (L149ff)
2. Added samples from texture-shape cue-conflict (Fig. 9 of the appendix)
3. Added an analysis of agreement sets, i.e., samples where all models predict shape, texture, a correct, or wrong label  (Appendix I).

---

### Meta-Review · Area_Chair_QGNn · 2024-12-20

**Metareview:**

Paper presents an approach that studies shape vs. texture bias in VLM models. Further, it studies steering of visual biases through language prompts, allowing shape bias to be dynamically adjusted. The paper was reviewed by four expert reviewers and received: 2 x accept, good paper and 2 x marginally above the acceptance threshold ratings. All reviewers acknowledge that the paper is well written, has clear motivation, and the analysis comprehensive. Main concerns and comments focused on (1) additional analysis, (2) additional qualitative results, (3) narrow scope of analysis (which excludes attention, etc.), (4) limited discussion on prompt selection, and (5) exposition and motivation with respect to prompt steering. Authors have provided a rebuttal that has addressed most reviewer comments with [VmoU] and [1KNs] rasing their scores. Overall, there appears to be consensus among the reviewers that this a solid paper with interesting insights and relatively through experimentation (with some additional results provided in the rebuttal).

AC has read the reviews, rebuttal, discussion that followed and the paper itself. AC agrees with consensus of the reviewers and is recommending Acceptance. Authors are encouraged to incorporate rebuttal responses and experiments either into the main paper or into the supplementals.

**Additional Comments On Reviewer Discussion:**

Authors have provided a rebuttal that has addressed most reviewer comments. Specifically, reviewer [VmoU] states that "concerns have been addressed in the author's rebuttal, and I will increase my scores"; reviewer [Ume1] mentions that he/she "have no other questions" and reviewer [1KNs] acknowledges that concerns are resolved" and that will lead to "raise [of] scores". Unfortunately, [bGZu] did not respond to the rebuttal. Overall, the positive responses of the reviewers post-rebuttal have lead to the recommendation mentioned above.

---

### Decision · Program_Chairs · 2025-01-22

Accept (Poster)